# CRaSh: Clustering, Removing, and Sharing Enhance Fine-tuning without Full Large Language Model

**Kaiyan Zhang[1], Ning Ding[1,2], Biqing Qi[1,2,3], Xuekai Zhu[1]**
**Xinwei Long[1], Bowen Zhou[1,2*]**

[1] Department of Electronic Engineering, Tsinghua University, Beijing, China
[2] Frontis.AI, Beijing, China
[3] School of Astronautics, Harbin Institute of Technology, Harbin, China
zhang-ky22@mails.tsinghua.edu.cn
zhoubowen@tsinghua.edu.cn

## Abstract

Instruction tuning has recently been recognized as an effective way of aligning Large Language Models (LLMs) to enhance their generalization ability across various tasks. However, when tuning publicly accessible, centralized LLMs with private instruction data, privacy concerns are inevitable. While direct transfer of parameterized modules between models is a plausible approach to address this, its implications and effectiveness need further exploration. This paper focuses on Offsite-Tuning (OFT), a representative technique that transfers transformer blocks between centralized LLMs and downstream emulators. Given the limited understanding of the underlying mechanism of OFT, we perform an empirical analysis on LLMs from the perspectives of representation and functional similarity. Interestingly, our findings reveal a unique modular structure within the layers of LLMs that appears to emerge as the model size expands. Simultaneously, we note subtle but potentially significant changes in representation and intermediate predictions across the layers. Inspired by these observations, we propose CRaSh, involving Clustering, Removing, and Sharing, a training-free strategy to derive improved emulators from LLMs. CRaSh significantly boosts performance of OFT with billions of parameters. Furthermore, we investigate the optimal solutions yielded by fine-tuning with and without full model through the lens of loss landscape. Our findings demonstrate a linear connectivity among these optima falling over the same basin, thereby highlighting the effectiveness of CRaSh and OFT. The source code is publicly available at https://github.com/TsinghuaC3I/CRaSh.

## 1 Introduction

Nowadays, there is a growing interest in large language models (LLMs) such as PaLM (Chowdhery et al., 2022), LLaMA (Touvron et al., 2023) and

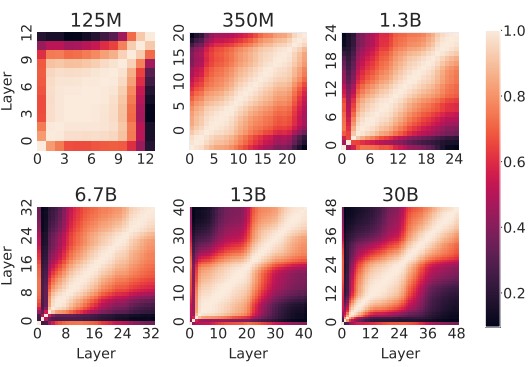

Figure 1: Emergence of modular structure of representations of LLMs. The table displays representation similarity among layers of OPT models (Zhang et al., 2022) on ARC dataset (Clark et al., 2018) with instruction format. The lighter colors indicate higher similarity and same color scale is used in all plots.

GPT-4 (OpenAI, 2023) due to their potential towards advanced intelligent systems. By employing techniques like prompt learning (Ding et al., 2022a; Wei et al., 2022b) and instruction tuning (Sanh et al., 2022; Wei et al., 2022a; Wang et al., 2022b,c), the behavior of LLMs can be aligned with human intent using a small amount of data. (Taori et al., 2023; Chiang et al., 2023; Zhou et al., 2023).

However, the centralization of LLMs poses a significant challenge concerning the trade-off between high performance and user data (Li et al., 2022a). For instance, OpenAI provides fine-tuning APIs[*] that allows users to upload personal data for further fine-tuning of davinci models family, which has gained popularity in the industry, like OpenAI GPT-4 (OpenAI, 2023), Google Bard[†], and Anthropic Claude[‡]. Safeguarding the privacy of both LLMs and downstream user data is an urgent concern. One viable approach involves the direct transfer of parameterized modules between models, such as Federated Learning (FL) (McMahan

---

[*]Corresponding author

[*]https://platform.openai.com/docs/guides
[†]https://bard.google.com/
[‡]https://www.anthropic.com/product

et al., 2017; Lin et al., 2022) and Split Learning (SL) (Vepakomma et al., 2018; Thapa et al., 2022), where limited exploration are conducted on LLMs with billions of parameters. Recently, Xiao et al. (2023) propose Offsite-Tuning (OFT) for transfer learning that operates independently of full LLMs with sizes exceeding 1B. OFT entails compressing the LLM into a smaller model known as emulator by layer dropping, followed by fine-tuning the emulator using user data. Finally, the parameterized modules are transferred from emulator and seamlessly integrated into LLM in a single turn. Despite the promising results obtained by OFT, there is still limited understanding of its underlying mechanism.

In this paper, we conduct a detailed analysis of LLMs from the perspective of representation and functional similarity (Kornblith et al., 2019; Belrose et al., 2023) to enhance our understanding of OFT. Through comparing the similarity of hidden states across layers, we observe *emergence of modular structure* of representations within LLMs. As depicted in Figure 1, models with a size less than 10B exhibit uniform representations across all layers. However, for models of size 13B and 30B, modular structure becomes apparent in the representation similarities. For the 13B model, high similarities are observed between layers 5 and 20, forming a light-colored block-diagonal structure (i.e., *modular structure*), as well as between layers 25 and 40. Additionally, we note subtle changes in the representation between adjacent layers. We further analyze functional similarity to explore the intermediate predictions of each layer, which enhances these findings.

Building upon our findings, we propose a completely training-free strategy to enhance fine-tuning without relying on full model. This strategy consists of three steps, namely Clustering, Removing, and Sharing (CRaSh). In the initial step, we cluster adjacent layers based on their similarity using various hierarchical clustering algorithms (Murtagh and Contreras, 2012). We then remove layers within the same cluster to obtain an emulator. The remaining layers are shared as a supplement to the removed layers. Finally, selected layers of the emulator are updated using downstream data and transferred to be seamlessly integrated into LLMs, resulting in improved performance. Through the utilization of CRaSh, we significantly enhance the performance of fine-tuning LLMs without full models across multiple datasets. In order to comprehend

the relationship of optimal solutions in CRaSh, we visualize the optima using loss surfaces and mode connectivity (Li et al., 2018; Frankle et al., 2020). This study offers valuable insights into the effectiveness of CRaSh. In summary, our main contributions can be summarized as follows:

- We discover emergence of *modular structure* within layers along with size of model increase in decoder-only models (e.g., OPT and LLaMA), which shows clusters of similar representations across layers (Section. 2).

- We propose CRaSh, a training-free strategy to enhance layer dropping compression (Section. 3). CRaSh improves the performance of fine-tuning without full model and even outperforms knowledge distillation on multiple datasets (Section. 4).

- We analyze the optima of fine-tuning with and without full model through the lens of loss surface and mode connectivity, which shows the two minima fall over the same basin and are connected linearly in parameter space. This observation explains the effectiveness of CRaSh (Section. 5).

## 2 Empirical Analysis

In this section, we analyze the inherent similarity of layers in LLMs from two complementary perspectives: (i) representation similarity (Section. 2.1), which examines the differences in activations among intermediate layers, and (ii) functional similarity (Section. 2.2), specifically the variations in predictions among intermediate layers.

### 2.1 Representation Similarity

Given a LLM, which produces word representations (i.e., hidden states) at each layer and aggregates them into sentence representations. We measure the similarity of sentence representations among layers using linear centered kernel alignment (CKA) (Kornblith et al., 2019). CKA emphasizes the distributivity of information. If two layers exhibit similarity across all their neurons, the similarity will be higher, even if individual neurons do not have similar matching pairs or are not well represented by all neurons in the other layer. The representation similarity reveals correlations between layers within LLMs. The inherent similarity in the representation of layers across two

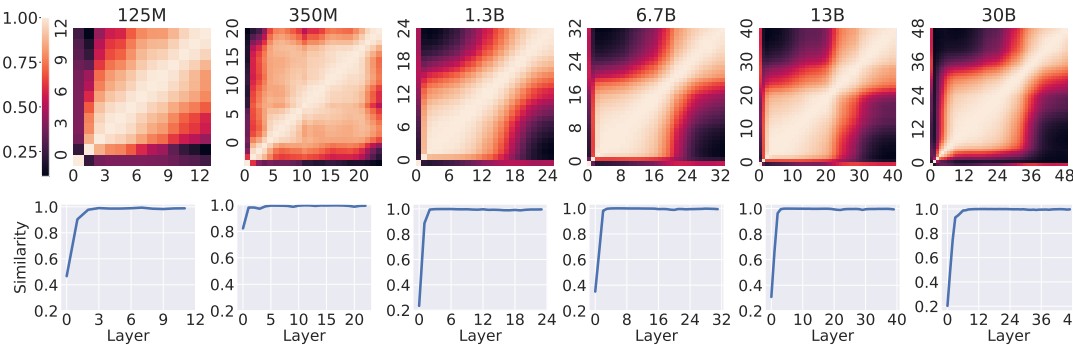

Figure 2: This table presents the representation similarity among all layers (shown above the heatmaps) and the similarity between adjacent layers (represented by the line charts) on the Wikitext corpus (Merity et al., 2017).

datasets are demonstrated in Figure 1 and Figure 2. We provide a detailed explanation of our two main findings as follows:

***Finding 1: Emergence of modular structure.*** As shown in Figure 1 and Figure 2, we observe that as the number of parameters increases, distinct blocks or modules with high similarity emerge in the representations of different layers. We refer to these blocks of high-similarity representations as *modular structure*. The emergence of *modular structure* can be seen as a self-organizing behavior during the pre-training of LLMs, where the internal representations gradually differentiate into modules with specific functions or semantics. This phenomenon has been briefly examined in previous studies (Phang et al., 2021; Merchant et al., 2020; Chen et al., 2021; Chiang et al., 2020). To the best of our knowledge, no study has investigated this question on LLMs with billions of parameters.

Additionally, our initial findings indicate that there are no evident clusters of layers in pre-trained models with millions of parameters, which may indicate insufficient capacity to comprehend tasks in zero-shot setting. However, when the model size reaches a certain threshold, referred to as *modular point*, *modular structure* emerges in LLMs, resulting in the formation of distinct clusters. Our experiments have shown that the specific *modular point* varies across tasks. Furthermore, we observe that the *modular point* is larger for harder tasks, but smaller for easier tasks, such as language modeling. For example, in the case of the ARC dataset (a question answering task with instruction format), *modular point* is observed to be 10B, as depicted in Figure 1. On the other hand, for the Wikitext corpus (a language modeling task), *modular point* is found to be 1.3B, as illustrated in Figure 2.

***Finding 2: Subtle changes in representation***

***among layers.*** In addition to *modular structure*, we can also observe high similarity along the diagonal of the similarity matrix. This indicates that intermediate layers may exhibit significant similarity to their adjacent layers. We directly examine the representation similarity between adjacent layers in Figure 2, where point at $x_i$ means similarity between layer $i$ and layer $i + 1$. In addition to the low similarity observed in the bottom layers, starting from about the 5th layer, there is a significantly higher similarity compared to adjacent layers.

Furthermore, we broaden the scope of these findings to include more LLMs and datasets, uncovering a consistent trend of *modular structure* emergence. The details of results, as well as implementation details, are provided in Appendix A.1.

## 2.2 Functional Similarity

In addition, a complementary perspective to representation similarity involves extracting specific concepts from the hidden states. For example, we can convert the hidden states at each intermediate layer into a probability distribution over the vocabulary (nostalgebraist, 2020; Belrose et al., 2023). This approach facilitates a deeper comprehension of the functional aspects of the layers. For the analysis of functional similarity, we utilize the state-of-the-art tool Tuned-Lens[§] to obtain predictions from the intermediate layers. Additional details are provided in Appendix A.2.

***Finding 3: Removing layers within a cluster maximizes behavior invariance.*** In the sub-figure labeled ①full model of Figure 3, we observe subtle changes as the depth increases and note that adjacent layers display a high similarity in hidden predictions, resulting in the formation of clusters.

---

[§]https://github.com/AlignmentResearch/tuned-lens

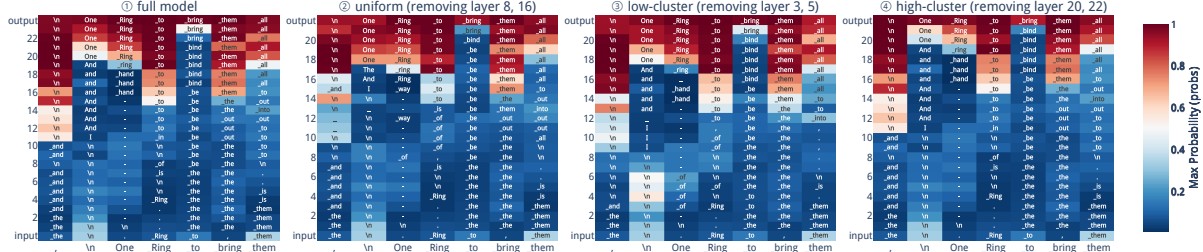

Figure 3: This figure shows results of Tuned-lens on OPT-1.3B and three variants of layer dropping. The entire input text consists of *"One Ring to rule them all,\n One Ring to find them,\n One Ring to bring them all\n and in the darkness bind them"*, where tokens from position 14 to 21 are indicated.

To assess the redundancy of layers, we eliminate multiple layers from the cluster based on representation similarity in Figure 2 and compare the predictions when these layers are uniformly dropped. Based on the results, we also observe all the removal strategies yield predictions that exhibit overall similarity to the internal layers of the full model, showing only minor differences in probability, as indicated by subtle variations in color shades. In comparison to the ①full model, the ②uniform strategy exhibits notable differences in the middle layers of token one, high layers of token four, and the last two tokens, indicating the presence of impure predictions in these positions. Conversely, the ③low-cluster and ④high-cluster strategies display closer alignment with the full model at these locations. This phenomenon leads to the conclusion that adjacent layers with high representation similarity also demonstrate analogous functional similarity. Furthermore, removing layers from these clusters results in maximum behavior invariance compared to uniformly dropping them.

## 3 Methodology

Empirical analysis of LLMs can inspire fine-tuning approaches, contributing to privacy protection for both centralized LLMs and user data. In this section, we begin by revisiting Offsite-Tuning (OFT), a representative method for fine-tuning LLMs without full model. Subsequently, we introduce our proposed CRaSh method inspired by Section. 2.

### 3.1 Revisit Offsite-Tuning

In this section, we provide a brief introduction to the settings and evaluation metrics of Offsite-Tuning (OFT) (Xiao et al., 2023). The primary concern of OFT is to maintain the privacy of both the LLM and user data. Specifically, *the data owner is unable to share their labeled training data with the*

*LLM owner, and vice versa, the LLM owner cannot share their model with the data owner.*

As shown in Figure 4a, OFT involves compressing the LLM into a smaller emulator by layer dropping and fine-tuning it using privacy data (i.e., ③Emulator Fine-tuning). Subsequently, the block weights are transferred and plug-in the LLM for inference (i.e., ④Plug-in). The main objective of OFT is for the plug-in to exceed the performance of both full zero-shot on the LLM and fine-tuning on the emulator (④ > ①, ③), thereby ensuring the overall effectiveness of OFT. Additionally, OFT aims to optimize the performance of the plug-in to closely approximate the results achieved by full fine-tuning on the LLM (④ ≈ ②).

The key to OFT lies in identifying an emulator that closely resembles the original LLM and subsequently performing fine-tuning to replace the LLM. Drawing from our findings on the empirical analysis of LLMs in Section 2, we propose CRaSh, which is designed to optimize layer dropping and yield superior sub-layer emulators from LLMs, as illustrated in the following sections.

### 3.2 CRaSh

In this section, we provide a detailed description of the three steps of CRaSh, namely Clustering, Removing, and Sharing as presented in Figure 4b. The concept of Clustering and Sharing are supported by *Finding 1*, which suggests that layers within a cluster share similar functions. Additionally, *Finding 2* and *Finding 3* support the idea of removing layers from clusters, as it helps minimize changes in representations and functions.

**Clustering** Drawing inspiration from the presence of layer clusters in representations and the gradual change in layer behavior in Section. 2, it is observed that adjacent layers within a cluster may have similar functions. These layers can be

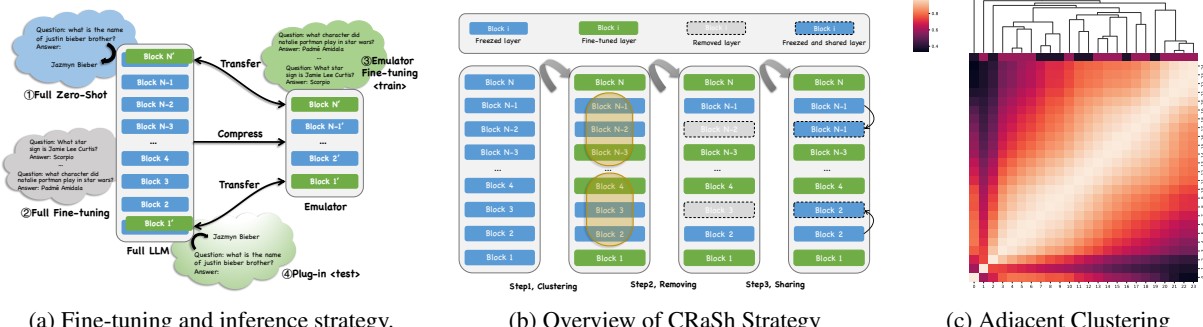

|                                      |                                    |                              |
|--------------------------------------|------------------------------------|------------------------------|
| (a) Fine-tuning and inference strategy. | (b) Overview of CRaSh Strategy  | (c) Adjacent Clustering      |

Figure 4: Overview of Offsite-Tuning and CRaSh strategy.

grouped together into a single cluster and subsequently replaced by the cluster center. As shown in Figure 4c, we propose a process for clustering the layers of LLMs based on the similarity of representations between adjacent layers. Initially, each individual layer is considered as a separate cluster. Subsequently, the closest clusters are selected and merged into a new cluster based on the CKA metric of their output representations. This step is repeated until the desired number of clusters is reached. This process bears resemblance to hierarchical clustering (Murtagh and Contreras, 2012), the key distinction lies in the fact that we exclusively cluster the adjacent layers.

**Removing** Following the clustering of intermediate layers, only layers located at the cluster centers are retained, while the rest within the clusters are removed. Subsequently, a selection of layers is chosen for fine-tuning, where parameters of the bottom and top $n$ layers are learnable in OFT (Xiao et al., 2023) (with $n = 2$). Considering the significance of knowledge contained in the middle layers for downstream tasks, we uniformly select a set of $n$ layers from the remaining layers after removal. The exploration of skill layers (Sajjad et al., 2023; Jordao et al., 2023) based on module cruciality is left as a potential area for future research.

**Sharing** In the original OFT approach, the model with the remaining layers is considered as an emulator and utilized for fine-tuning on downstream user data. This allows users to experiment with different strategies to optimize the emulator's performance. For downstream fine-tuning, we propose a straightforward strategy to enhance the performance of emulator. Considering the emergent abilities of LLMs, certain features, such as chain-of-thought (Wei et al., 2022b), may not be present in shallow layer models. Drawing inspiration from the concept of layer sharing (Lan et al., 2020),

we implement layer sharing within the remaining model to achieve optimal results.

## 4 Experiments

### 4.1 Setup

**Dataset.** We evaluate CRaSh on three tasks and eight datasets which are used in Offsite-Tuning (Xiao et al., 2023), including Multi-Choice QA: OpenBookQA (Mihaylov et al., 2018), PIQA (Bisk et al., 2020), SciQ (Welbl et al., 2017), RACE (Lai et al., 2017); Closed-Book QA: ARC-Easy/Challenge (Clark et al., 2018), WebQuestion (Berant et al., 2013); and Sentence Completion: HellaSwag (Zellers et al., 2019). To enhance zero-shot performance, we organize source and target text in instruction format as `lm-evaluation-harness` (Gao et al., 2021). We utilize it to evaluate our models and report the accuracy on all benchmarks. For clustering step, we utilize data from the same task as support dataset to maintain the privacy of target dataset, including BoolQ (Clark et al., 2019), TriviaQA (Joshi et al., 2017), and CoPA (Wang et al., 2019). We show details about statistic of datasets in Appendix B.1.

**Models.** We perform empirical analysis on a range of models, including OPT (from 125M to 30B) (Zhang et al., 2022) and LLaMA (from 7B to 30B) (Touvron et al., 2023). Due to limited computational resources, we primarily conduct main experiments on OPT-1.3b, with plans to scale up to OPT-6.7B and LLaMA-7B. As our primary baselines, we consider models from OFT (Xiao et al., 2023), including a knowledge distillation emulator and a uniform `2-8-2` configuration for OPT-1.3b, `2-18-2` for OPT-6.7B and LLaMA-7B. Here, `l-c-r` denotes setting the parameters of bottom $l$ and top $r$ layers as learnable, while $c$ layers are uniformly selected from original LLM and kept

| Setting | OpenBookQA | ARC-E | ARC-C | WebQs | PIQA | SciQ | RACE | HellaSwag |
|---|---|---|---|---|---|---|---|---|
| ***Full Large Language Model*** | | | | | | | | |
| Zero-shot (ZS) | 23.4% | 56.9% | 23.5% | 4.6% | 71.6% | 84.4% | 34.2% | 41.5% |
| Fine-tuning (FT) | 31.4% | 61.3% | 27.7% | 31.2% | 75.2% | 92.5% | 37.0% | 42.7% |
| ***Knowledge Distillation (Continual Training)*** | | | | | | | | |
| Emulator ZS | 19.4% | 53.9% | 21.5% | 1.3% | 68.7% | 80.9% | 33.0% | 35.1% |
| Emulator FT | 24.8% | 58.1% | 26.1% | 24.3% | 71.6% | 92.2% | 38.6% | 37.0% |
| Plug-in (Xiao et al., 2023) | **29.0%** | **59.4%** | **27.8%** | **26.2%** | **74.5%** | **92.9%** | **38.9%** | **43.3%** |
| ***Uniform Strategy (Training-free)*** | | | | | | | | |
| Emulator ZS | 13.8% | 34.9% | 19.0% | 0.0% | 58.4% | 49.8% | 22.7% | 27.0% |
| Emulator FT | 24.6% | 50.4% | 21.2% | 21.8% | 69.3% | 89.4% | 36.5% | 32.7% |
| Plug-in (base) | **26.4%** | **58.3%** | **23.0%** | **21.4%** | **72.7%** | **90.8%** | **37.9%** | **41.2%** |
| *with uniform learnable layers* | | | | | | | | |
| Emulator FT | 24.2% | 51.1% | 24.1% | 23.7% | 69.3% | 89.3% | 36.9% | 33.8% |
| Plug-in | 27.6% | 58.8% | 24.8% | 16.4% | 72.6% | 92.1% | 39.7% | 41.2% |
| ***CRaSh Strategy (Training-free)*** | | | | | | | | |
| Emulator ZS | 14.0% | 35.9% | 18.5% | 4.7% | 57.0% | 84.3% | 34.2% | 25.9% |
| Emulator FT | 25.0% | 50.0% | 21.5% | 21.8% | 68.9% | 88.9% | 38.9% | 33.6% |
| Plug-in (our) | **30.2%** ↑4.8 | **60.0%** ↑1.7 | **24.8%** ↑1.8 | **23.7%** ↑2.3 | **73.2%** ↑0.5 | **93.1%** ↑2.3 | **39.9%** ↑2.0 | **41.9%** ↑0.7 |
| *w/o layer sharing* | | | | | | | | |
| Emulator ZS | 14.0% | 35.9% | 18.5% | 0.0% | 57.0% | 43.3% | 23.6% | 25.9% |
| Emulator FT | 23.8% | 50.0% | 21.5% | 24.3% | 68.8% | 89.6% | 36.2% | 31.2% |
| Plug-in | 25.2% | 57.7% | 24.6% | 17.7% | 71.7% | 92.2% | 39.1% | 41.3% |

Table 1: Results of CRaSh on OPT-1.3B. The values in red font indicate an increase compared to the uniform strategy, highlighting the superiority of CRaSh. Remarkably, CRaSh outperforms the strategy with knowledge distillation (KD) on several datasets. It is worth noting that KD can further enhance the performance of CRaSh.

frozen during fine-tuning. We present implementation details about experiments in Appendix B.2.

## 4.2 Experimental results

We present the main results in Table 1 and provide an analysis as follows.

**OFT and CRaSh does work well.** CRaSh satisfies the condition of OFT, where the performance of plug-in is better than the zero-shot performance of full model and the fine-tuning performance of emulator. The results prove that OFT effectively works for fine-tuning without full model and is beneficial for protecting the privacy of LLMs. Additionally, due to over-parameterization of LLMs (Aghajanyan et al., 2021; Ding et al., 2022b), it may not be necessary to optimize all parameters. Therefore, the performance of plug-in can surpass that of directly fine-tuning on LLMs, particularly on datasets like SciQ and RACE.

**CRaSh is better than uniformly dropping startegy.** Compared to the uniform layer dropping strategy used in OFT (Xiao et al., 2023), CRaSh is an effective method to boost performance that requires low additional cost and is completely training-free, where lifting effects are indicated in red font. Additionally, CRaSh can outperform the knowledge distillation models in OFT settings on several datasets, such as achieving 1.2% improve-

ments on OpenBookQA and 1.0% on RACE. The knowledge distillation method obtains the emulator by continuously pre-training it on the first block of the Pile corpus (Gao et al., 2020), which aims to align the intermediate representation between emulator and full model. Therefore, CRaSh is complementary to knowledge distillation, which can further improve performance by leveraging the improved emulator initialization provided by CRaSh.

**CRaSh works better while scaling up model.** As depicted in Table 2, CRaSh remains effective even when scaling up the model from 1B to 7B. The benefits and effectiveness of CRaSh are not restricted to specific model sizes, making it a valuable strategy for enhancing OFT outcomes across models of varying scales. Meanwhile, as the depth increases, layers are more prone to redundancy in LLMs. Therefore, by employing a clustering step, CRaSh eliminates redundant network layers and effectively improves the performance of OFT compared to the uniform strategy.

## 4.3 Ablation study

**Impact of clustering and sharing steps.** This section discusses the significance of clustering and sharing steps in CRaSh. We compare the results with and without the clustering step, as shown in Table 1. In OFT (Xiao et al., 2023), the top and

| Setting | OpenBookQA | ARC-E | ARC-C | WebQs | PIQA | SciQ | RACE | HellaSwag |
|---|---|---|---|---|---|---|---|---|
| *OPT-6.7B* | | | | | | | | |
| Full ZS | 27.6% | 65.6% | 30.6% | 8.8% | 76.2% | 90.1% | 38.2% | 50.5% |
| Emulator ZS | 21.4% | 55.6% | 23.9% | 1.5% | 57.0% | 84.1% | 31.1% | 28.4% |
| Emulator FT | 29.0% | 60.1% | 31.1% | 22.1% | 75.6% | 88.4% | 36.2% | 43.4% |
| Plug-in (Xiao et al., 2023) | 33.8% | 66.8% | 33.9% | 23.9% | 77.7% | 91.9% | 44.1% | 52.1% |
| **Plug-in (CRaSH)** | **38.8%** $_{\uparrow 5.0}$ | **70.7%** $_{\uparrow 3.9}$ | **36.3%** $_{\uparrow 2.4}$ | **26.1%** $_{\uparrow 2.2}$ | **78.0%** $_{\uparrow 0.3}$ | **95.3%** $_{\uparrow 4.2}$ | **45.2%** $_{\uparrow 1.1}$ | **53.4%** $_{\uparrow 1.3}$ |
| *LLaMA-7B* | | | | | | | | |
| Full ZS | 28.2% | 67.3% | 38.2% | 0.0% | 78.3% | 89.7% | 40.0% | 56.4% |
| Emulator ZS | 15.0% | 44.3% | 23.5% | 0.0% | 65.7% | 57.6% | 30.2% | 36.2% |
| Emulator FT | 25.4% | 60.0% | 28.8% | 25.7% | 73.6% | 91.8% | 40.8% | 45.0% |
| Plug-in (Uniform) | 33.0% | 69.6% | 39.0% | 27.3% | 78.8% | 93.5% | 44.0% | 57.4% |
| **Plug-in (CRaSH)** | **34.6%** $_{\uparrow 1.6}$ | **71.3%** $_{\uparrow 1.7}$ | **41.8%** $_{\uparrow 2.8}$ | **29.8%** $_{\uparrow 2.5}$ | **80.0%** $_{\uparrow 1.2}$ | **95.1%** $_{\uparrow 1.6}$ | **45.6%** $_{\uparrow 1.6}$ | **58.4%** $_{\uparrow 1.0}$ |

Table 2: Results on OPT-6.7B and LLaMA-7B. CRaSh continues to perform effectively as the model size scales up, and even achieves further improvement, benefiting from the emergence of *modular structure*.

| Dataset | OpenBookQA | ARC-E | ARC-C | WebQs |
|---|---|---|---|---|
| The Wikitext | 27.8% | 58.5% | 24.0% | 22.0% |
| Support Task | 30.2% | 60.0% | 24.8% | 23.7% |
| Downstream Task | 29.4% | 59.6% | 25.1% | 24.1% |

Table 3: The table presents the plug-in results of CRaSh, considering different data types for the clustering step.

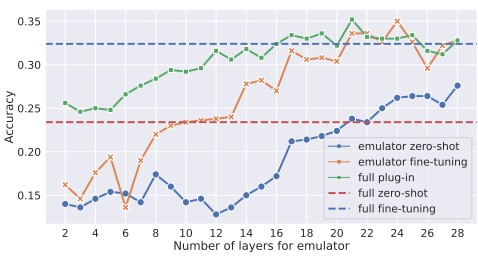

Figure 5: The accuracy varies as the number of layers to be dropped changes on OPT-6.7B model.

bottom two layers are set to be learnable, while the four layers to be updated are uniformly chosen in our CRaSh strategy. Hence, we run the *Uniform Strategy* in *"with uniform learnable layers"* and compare it with *CRaSh Strategy* without layer sharing (e.g., *"w/o layer sharing"*). In cases where only the layers of the emulator differ, we find that the clustering step provides slight utility compared to the uniform strategy. This trend may be attributed to the potential loss of hidden information caused by layer dropping. However, this loss can be mitigated by incorporating layer sharing and fine-tuning, resulting in improved performance of the final CRaSh approach. The results in Table 1 also indicate the significance of layer sharing steps, as they enable the emulator to have sufficient depth to handle challenging tasks.

**Impact of data type for clustering.** In our main experiments, we take into account the privacy of

downstream data. Therefore, we solely utilize data from the support task for clustering, which shares a similar task type with the downstream task. As presented in Table 3, we compare this approach with using public general task and directly downstream dataset in order to examine the impact of data type on clustering and the resulting plug-in performance. By avoiding direct use of the downstream task, which may compromise privacy, we achieve comparable performance by utilizing data solely from a similar support task. We leave it for future work to explore the identification of the most relevant support task for clustering based on downstream task information.

**Number of layers for emulator.** To ensure a fair comparison, we set the number of layers of emulator to 12 and 22 in main experiments, respectively, for 1B and 7B LLMs. To investigate the impact of an extensive range of layers, we varied the number of clusters from 2 to the maximum number of total layers. The performance of plug-in increases in accordance with the number of layers for the emulator, as depicted in Figure 5. Additionally, we observed that the performance of plug-in can achieve a comparable effect to full fine-tuning when the layers comprise only 50% ∼ 60% of the LLMs. However, there is still room for further exploration when transferring fewer layers of LLMs.

## 5 Discussion

**Loss landscape and mode connectivity.** In order to comprehend the effectiveness of CRaSh, we conduct an analysis of the relationship between optimal minima derived from plug-in and full model fine-tuning. Firstly, based on Figure 6a, we observe that the initialization resides in a low basin and per-

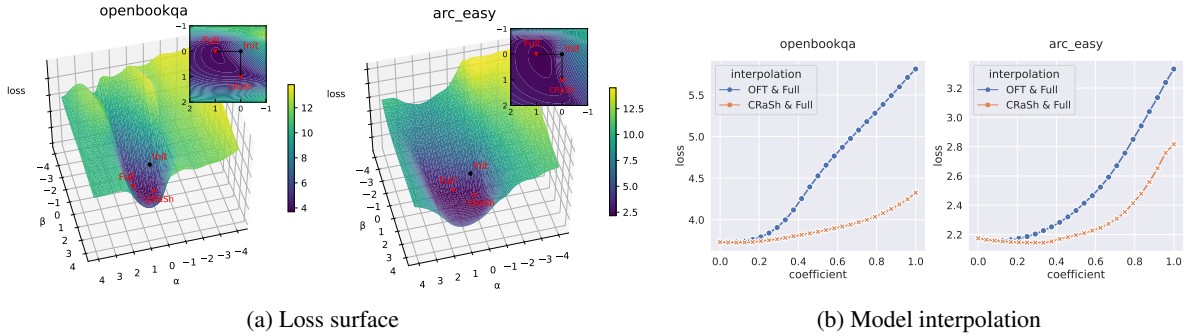

(a) Loss surface

(b) Model interpolation

Figure 6: (a) The initialization weights and optima obtained through CRaSh and full model fine-tuning are located within the same basin. (b) Model interpolation on weights from CRaSh, OFT, and full fine-tuing.

forms well across various datasets, which benefits from the over-parameterization and generalization of LLMs. Consequently, by optimizing the LLM at a relatively low cost on the target dataset, it can effectively perform on this dataset. This observation highlights the efficacy of delta-tuning (Ding et al., 2022b) and offsite-tuning. Secondly, the solutions derived from CRaSh and full fine-tuning reside within the same basin. Utilizing the clustering and sharing steps, CRaSh exhibits a closer proximity to full fine-tuning compared to OFT (refer to Figure 6b). Finally, through the interpolation of various solutions, we observe their mode connectivity in the parameter space, wherein CRaSh exhibits a smoother transition towards full fine-tuning compared to OFT, which undergoes more dramatic changes. Based on the visualization of the loss landscape, there is potential to enhance CRaSh for better performance in future directions.

**Paramter-Efficient Fine-tuning (PEFT).** PEFT has gained popularity as a method for adapting LLMs with minimal parameter updates or additions. For instance, in the case of LoRA, only 590K parameters need to be updated for OPT-1.3B, while OFT requires updating 201M parameters and the full model requires updating 1208M parameters. By applying LoRA to the transferred layers, CRaSh achieves parameter- and resource-efficient fine-tuning. Further details on this topic can be found in Appendix C.2.

**Reconstruct full model from emulator.** When transmitting the emulator downstream, an important consideration emerges: given the emulators, how challenging would it be to reconstruct the original model? The core question is the attainable performance level with just the emulator. Our findings, detailed in Section 4, discuss the complexities involved in reconstruction: (1) Layer sharing

in CRaSh is viewed as the most effective reconstruction technique. However, it does not replicate the performance of the original model entirely. As indicated in Table 2, the emulator fine-tuned with CRaSh does not achieve the full model's zero-shot performance. But, when integrated, there's a marked improvement in performance across multiple datasets, especially ARC-E/C, PIQA, RACE, and HellaSwag. (2) A crucial factor in the reconstruction challenge is the number of transferred layers; fewer transferred layers complicate the reconstruction. Figure 5 demonstrates that accuracy varies with the number of layers omitted. Interestingly, emulators with merely 6 layers (2 frozen and 4 fine-tuned, compared to the 32 layers in the full model) continue to boost the primary LLM's performance when incorporated. Undoubtedly, CRaSh could be enhanced by incorporating federated learning technologies, including homomorphic encryption (Lee et al., 2022; Chen et al., 2022) and differential privacy (Yue et al., 2021), making the original model's reconstruction more challenging.

## 6 Related Work

Delta-tuning (Ding et al., 2022b) methods efficiently update a subset of parameters compared to the entire LLMs (Houlsby et al., 2019; Ben Zaken et al., 2022; Hu et al., 2022; Dettmers et al., 2023). However, these methods require feeding data into the entire LLMs, which is not resource-efficient and raises concerns about the privacy of LLMs. Black-box tuning (Sun et al., 2022b,a), as applied to centralized LLMs, attempts to learn parameters based on input or output text (Cui et al., 2022), which helps protect LLMs but poses risks to user data. Directly manipulating model parameters instead of transferring data has found wide applications in federated learning (McMa-

han et al., 2017), split learning (Vepakomma et al., 2018; Thapa et al., 2022), distributed gradient descent (Huo et al., 2018; Xu et al., 2020; Ni et al., 2023), branch-train-interpolation (Li et al., 2022b; Wortsman et al., 2022), and collaborative machine learning development (Raffel, 2023; Kandpal et al., 2023). These methods either involve model sharing between servers and clients or require multi-turn communication to achieve coverage, which poses limitations when applied to LLMs with billions of parameters. Offsite-Tuning (Xiao et al., 2023) is a notable method that addresses these challenges by selectively sharing layers of LLMs with billions of parameters between servers and clients in a single turn. On the other hand, previous studies have focused on gaining insights into the intermediate representation of neural networks for better fine-tuning (Kornblith et al., 2019; Merchant et al., 2020; Wu et al., 2020; Raghu et al., 2021). In terms of transformers, Phang et al. (2021) observed clustering of layer representations in fine-tuned BERT models(Devlin et al., 2019), supporting the notion of layer redundancy (Dalvi et al., 2020). In contrast, we find that clustering also emerges in pre-trained LLMs as the model size increases, which helps Offsite-Tuning on LLMs. A detailed introduction to related work is presented in Appendix D.

## 7 Conclusion

In this paper, we uncover block structure of representation within the intermediate layers of LLMs, indicating the clustering of layers in depth models. Based on these observations, we propose a completely training-free strategy to enhance fine-tuning without full model. The strategy consists of three steps: Clustering, Removing, and Sharing (CRaSh). CRaSh boosts the performance of Offsite-Tuning for LLMs on various datasets. Further analysis of the loss surface and mode connectivity provides insights into the effectiveness of CRaSh.

## Limitations

This paper primarily focuses on fine-tuning LLMs without using the full model, thereby safeguarding the privacy of both centralized model and data. Our empirical analysis on LLMs has inspired a straightforward yet effective improvement that outperforms previous methods.

However, We have only conducted experiments using two types of LLMs (OPT and LLAMA), leaving a wide range of other LLMs unexplored (such as Pythia). Due to limited computing resources, our main experiments were conducted on LLMs with a model size of less than 10B.

Although we hypothesize that larger models with redundant layers and CRaSh may lead to improved performance, further exploration is necessary to validate this hypothesis. We utilize representation similarity as a clustering metric, and although it demonstrates satisfactory performance in our experiments, we have encountered challenges and observed instability. Consequently, we intend to investigate functional similarity as an alternative, which may necessitate additional preprocessing time.

Given the complexity involved in validating the plug-in, further research on behavior predicting is required to enhance CRaSh (which tends to be a heuristic strategy) and transform it into an automated learnable strategy (such as reinforcement learning). For future work, we plan to do analysis using more similarity methods and broaden the application of this strategy to encompass a wider range of instructional data and LLMs.

## Ethics Statement

Our research on Offsite-Tuning (OFT) and the proposed CRaSh strategy aligns with the ethical guidelines outlined by the ACL Ethics Policy. We recognize the importance of addressing privacy concerns when fine-tuning publicly accessible, centralized Large Language Models (LLMs) with private instruction data.

The primary objective of our work is to enhance the generalization ability of LLMs across various tasks while safeguarding the privacy of both the LLMs and the instruction data. We acknowledge the potential risks associated with the direct transfer of parameterized modules between models and the need for further exploration to fully understand its implications and effectiveness.

Our research adheres to ethical principles by promoting privacy protection, transparency, and responsible use of LLMs. We are committed to continuous ethical evaluation and will contribute to the ongoing discourse on the ethical implications of language model fine-tuning.

## Acknowledgements

This work is supported by the National Key R&D Program of China (No. 2022ZD0119101). We extend our gratitude to the anonymous reviewers for their insightful feedback.

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

## A  Empirical Analysis

### A.1  Representation Similarity Details

To compute the similarity of representations between pairs of layers within LLMs, we employ the technique known as Centered Kernel Analysis (CKA), as introduced by Kornblith et al. (2019). For every layer within LLMs, the hidden state outputs encompass characteristics derived from the input tokens with the shape of $(N, L, H)$, denoted as $\mathbf{h}_i$. Here, $N$ signifies the batch size, $L$ denotes the input length, and $H$ represents the hidden size. After getting features of all input tokens, a pooling strategy is used to get sentence embedding for similarity computing, where max pooling is taken. We evaluate CKA between each pair of layers in the LLM to be compared. For layer $i$ and $j$, which denoted as $\mathbf{s}_i$ and $\mathbf{s}_j$, both with the shape of $(N, H)$, the linear CKA is given by:

$$\text{CKA}(\mathbf{s}_i, \mathbf{s}_j) = \frac{||\mathbf{s}_i^T \mathbf{s}_j||_F^2}{||\mathbf{s}_j^T \mathbf{s}_j||_F ||\mathbf{s}_i^T \mathbf{s}_i||_F}$$

where $|| \cdot ||_F$ denotes the Frobenius norm.

In contrast to BERT-style models, which incorporate a CLS token for sentence embedding, GPT-style models focus solely on preceding tokens. Therefore, the last token within GPT models often holds the most significant semantic representation. Nevertheless, according to the findings in the experimental study by Muennighoff (2022), employing the weighted mean pooling strategy, which assigns more importance to later tokens, leads to superior results in sentence-level semantic expression and ensures stability even as the depth of layers increases. Consequently, we adopt the weighted mean pooling strategy to obtain the sentence vector within the hidden layers:

$$R_i = \sum_{i=k}^{L} w_k \mathbf{h}_k, \; w_k = \frac{i}{\sum_{k=1}^{L} k}$$

where $\mathbf{h}_k$ is the $k$th hidden state and $R$ means the sentence embedding of $i$th layer.

We organize the source and target text samples into an instruction format, as illustrated in Table 5 and Table 6.

For CKA, representation similarity is computed across all input samples. Due to computational constraints, we use a set of 512 samples for computation, which has been found to be sufficient and stable for analyzing the modular structure.

Additionally, we provide results on more datasets and models in Figure 11, Figure 12, and Figure 15. While we employ weighted mean pooling to obtain sentence representations, we also include results using mean pooling in Figure 13 and Figure 14.

### A.2  Functional Similarity Details

Logits-Lens was proposed by (nostalgebraist, 2020) as a method to gain insights into the internal workings of GPT-2, with a specific focus on analyzing the logits, which represent the raw outputs generated by the model before applying the softmax function to obtain probabilities. Logits-Lens aims to examine the logits at various layers of GPT models in order to enhance understanding of prediction process in LLMs. Through the inspection of these logits, researchers and developers can gain valuable insights into decision-making process of LLMs and potentially discover underlying patterns or biases. Due to the instability of Logits-Lens, which fails to function effectively in larger and deeper models, (Belrose et al., 2023) introduces Tuned-Lens, where each layer of the model is trained using an affine transformation on a pre-training corpus.

## B  Main Experimental Details

### B.1  Dataset Details

We present statistical results of both downstream tasks (target tasks) and support tasks in Table 4.

### B.2  Implementation Details

For empirical analysis, we randomly select 512 samples from the evaluation set of each dataset for CKA computation. Additionally, we perform the clustering step by considering only layers between the $l$th bottom layers and the last $r$th top layers, taking into account the coarse changes among them. However, we ensure that the layers of the emulator remain the same as in Offsite-Tuning. We perform a learning rate tuning process on a grid of values and report the runs with the highest emulator performance, where $\{1e-4, 2e-4, 3e-4, 2e-5, 5e-5, 8e-5\}$ for OPT-1.3B and $\{5e-6, 8e-6, 1e-5, 2e-5, 5e-5\}$ for OPT-6.7B and LLaMA-7B. We also select the best adapter layers in both settings, with and without sharing, indicating that the sharing step is an optional component for local clients. Furthermore, the decision to repeat fine-tuned layers is dependent on the available local resources. The experiments

| Task | Multi-Choice QA | | | | | Closed-Book QA | | | | SentComp | |
|---|---|---|---|---|---|---|---|---|---|---|---|
| Dataset
Domain | OpenBookQA
Sci.Edu | PIQA
Physical | SciQ
Sci.Edu | RACE
Edu.Exam | BoolQ
Gen. | ARC-E
Gen. | ARC-C
Sci.Edu | WebQs
Gen. | TriviaQA
Gen. | HellaSwag
Gen. | CoPA
Gen. |
| Train.Size | 4,957 | 16,113 | 11,679 | 62,445 | 9,427 | 2,251 | 1,119 | 3,589 | 87,622 | 39,905 | 400 |
| Eval.Size | 500 | 1,838 | 1,000 | 3,451 | 3,270 | 570 | 299 | 189 | 11,313 | 10,042 | 100 |
| Test.Size | 500 | 3,084 | 1,000 | 3,498 | 3,245 | 2,376 | 1,172 | 2,032 | 10,832 | 10,003 | 500 |
| Avg.Context | 13.10 | 14.36 | 110.27 | 407.90 | 143.96 | 28.02 | 31.15 | 15.54 | 24.23 | 49.18 | 8.68 |
| Avg.Target | 4.87 | 24.28 | 3.68 | 8.68 | 2.0 | 5.93 | 7.28 | 4.24 | 4.48 | 29.52 | 7.34 |
| Avg.Total | 17.97 | 38.64 | 113.95 | 416.58 | 145.96 | 33.95 | 38.43 | 19.78 | 28.71 | 78.69 | 16.03 |

Table 4: Statistics are collected for all datasets, with the last one of each task serving as a support task for clustering (e.g., BoolQ, TriviaQA, and CoPA). Abbreviations: SentComp = Sentence Completion, Sci.Edu = Science Enducation, Edu.Exam = Enducation and Examination, Gen. = General domains.

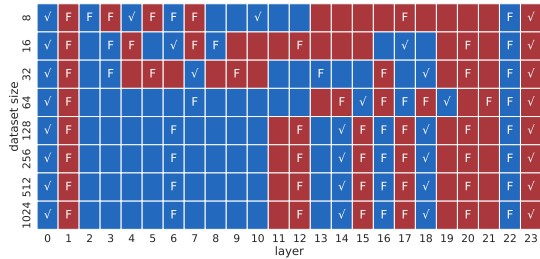

Figure 7: This figure presents clustering results on OPT-1.3B with the Wikitext Corpus, showcasing the influence of varying example numbers (8 to 1024). Adjacent layers within the same cluster are color-coded (red or blue). Emulator layers are marked with F and $\sqrt{}$, where the latter indicates learnable.

are conducted using the NVIDIA RTX 3090 GPUs for OPT-1.3B and A6000 GPUs for OPT-6.7B and LLaMA-7B.

# C Additional Experiments and Analysis

## C.1 Impact of data size for clustering

As illustrated in Section. 3.2, the CKA metric used in clustering step is computed based on the similarity of representation in intermediate layers.

According to previous studies (Wu et al., 2020; Csiszárik et al., 2021; Phang et al., 2021), increasing the amount of data used results in higher accuracy for the CKA metric. However, due to limited computing resources, it is challenging to load all examples into memory once for computation. To reduce memory consumption during computation, Nguyen et al. (2021) proposed computing linear CKA by averaging HSIC scores based on minibatches. However, being limited with computing resource, it's hard to feed all examples into memory for computing. In contrast to their approach, we focus on the clustering process itself and analyze the results using varying sizes of examples, as shown in Figure 7. Our findings indicate that the

clustering results tend to stabilize as the data size increases up to 128. Based on these findings, downstream users can now upload only a small amount of data relevant to their task, resulting in improved clustering.

## C.2 Combined with parameter-efficient tuning

OFT is independent of parameter-efficient tuning methods, including Adapter (Houlsby et al., 2019), BitFit (Ben Zaken et al., 2022), and LoRA (Hu et al., 2022). For example, in the case of LoRA, only 590K parameters need to be updated for OPT-1.3B, compared to 201M for OFT and 1208M for the full model. This demonstrates the parameter- and resource-efficiency achieved through OFT. Based on results presented in Figure 8, we observe that CRaSh is also compatible with LoRA and exhibits superior performance. Nevertheless, there is still a noticeable performance decline on various datasets that requires further exploration in future studies.

## C.3 Sharing among different layers

Layer sharing is a beneficial approach to enhance model capacity when faced with limited resources (Lan et al., 2020). The downstream user has the option to either share weights among layers or not, depending on the emulator's performance. In order to investigate the impact of weight sharing across different layers, we substitute each layer in LLMs with another layer. As illustrated in Figure 9, the heatmap above displays the loss on the validation dataset when replacing layer $i$ with layer $j$. The diagonal represents the original LLMs, where the loss is the lowest, and locations near the diagonal also exhibit low loss. Additionally, in the heatmap below, we compare the representations from the last layer of the original LLM and the layer sharing models, which demonstrates a similar

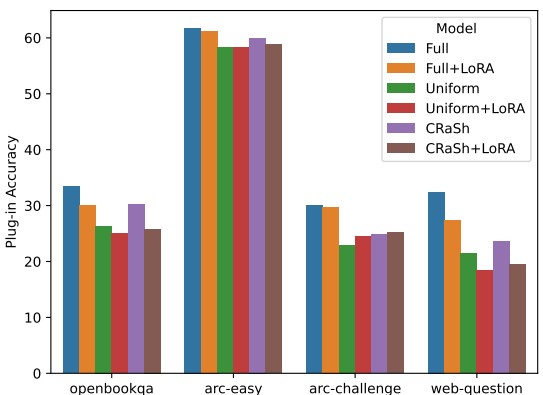

Figure 8: The table presents the plug-in accuracy for four datasets when combining LoRA with different strategies. It is evident that CRaSh performs well with LoRA compared to the uniform strategy. Although LoRA significantly reduces the number of learnable parameters by optimizing the model in a low-rank space, it negatively impacts performance in almost all datasets.

phenomenon. These analyses collectively indicate that weights sharing between adjacent layers can maintain comparable performance to the primary model. This finding supports the effectiveness of the layer sharing steps in CRaSh.

## C.4 Comparing layer clustering to module cruciality

As demonstrated in Figure 10, we showcase module cruciality through the process of rewinding and removing layers. The findings reveal that the majority of important neurons are concentrated in the middle and high layers of the model. This supports our decision to uniformly select learnable layers. However, further investigation is required for a more comprehensive understanding of this phenomenon.

## D Related Works

In this section, we first review the adaptation methods for LLMs, then introduce a technique called fine-tuning without full models. After that, we discuss the compression methods for LLMs. Finally, we present similarity methods for interpreting LLMs to enhance adaptation.

**Parameter- and resource-efficient fine-tuning** has emerged as a popular method for adapting LLMs. Instead of directly updating pre-trained models for downstream tasks, these methods focus on updating or adding a minimal number of parameters, sometimes not requiring the involvement of the entire parameter set. Regarding parameter-

efficient fine-tuning, Houlsby et al. (2019) introduced task-specific modules called adapters, such as bottleneck networks, into LLMs. Taking into account the intrinsic space of LLMs (Aghajanyan et al., 2021), LoRA (Hu et al., 2022) optimizes LLMs in a low-rank space, where additional matrices are updated and can be directly added to the original parameter matrix. Except for inserting new parameters, two other approaches, BitFit (Ben Zaken et al., 2022) and skill neurons (Wang et al., 2022a), aim to identify task-related parameters within LLMs and update them to incorporate task-specific knowledge while keeping the remaining model parameters frozen. Another approach is black-box tuning (BBT) (Sun et al., 2022b,a), which involves LLMs that are inaccessible to users. In this method, the continuous prompt prepended to the input text is optimized using derivative-free optimization techniques. However, in these methods, the full models are required to participate in forward propagation, and they do not provide any assistance in protecting the privacy of LLMs and data. Another approach is resource-efficient fine-tuning, which involves evolving only a subset of parameters during the fine-tuning process (Sung et al., 2022; Chen et al., 2023). These methods solely focus on transferring knowledge in one direction to downstream tasks and do not facilitate the transfer of updated weights back to the original model.

**Fine-tuning without using full models** is motivated by the need for privacy protection, which is not adequately addressed by current transfer learning methods. Federated learning (FL) (McMahan et al., 2017) involves distinguishing server and client models, where training is performed locally on the client and then aggregated. However, the client and server retain the same model, which compromises the privacy of LLMs. Another approach is decoupled learning, which decomposes the end-to-end optimization problem of neural training into smaller subproblems. This objective is achieved through various techniques, such as distributed gradient descent (Huo et al., 2018; Xu et al., 2020), model assembly (Ni et al., 2023), branch-train-interpolation (Li et al., 2022b; Wortsman et al., 2022), and collaborative development of machine learning (Raffel, 2023; Kandpal et al., 2023). However, these methods primarily focus on training from the beginning rather than during the fine-tuning phase or risk leaking the full model. To ad-

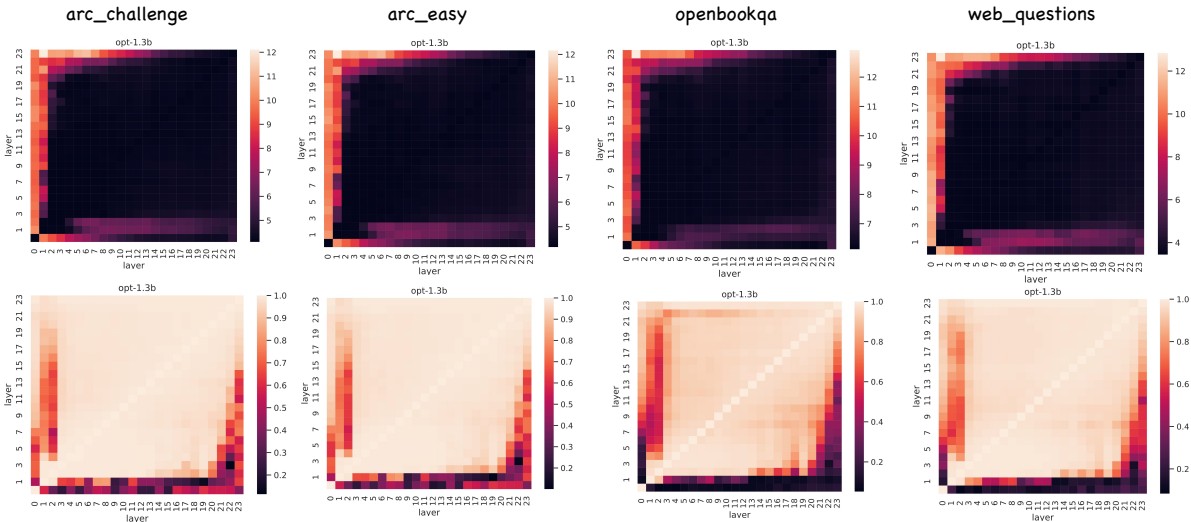

Figure 9: Analysis on layer sharing. $(i, j)$ means replacing layer $i$ with layer $j$, i.e. sharing layer $j$ with layer $i$. Above: loss on validation set. Below: representation similarity compared to last layer.

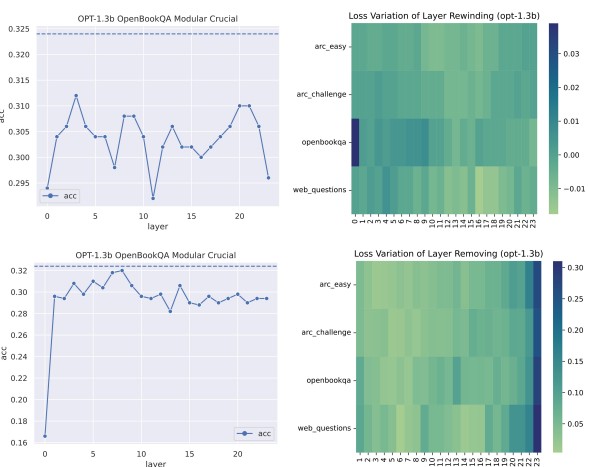

Figure 10: Given a full finetuned model, we evaluate specific layer importance via rewinding weights of layer to initialization (above) and directly removing layer from finetuned model (below).

dress this issue, Xiao et al. (2023) propose Offsite-Tuning, a novel approach that involves transferring compressed versions of LLMs to downstream tasks and performing fine-tuning on privacy-sensitive user data. The updated weights are then transferred back and incorporated into LLMs. This method shows promise for protecting the privacy of LLMs and user data. This work is highly relevant to our research, and we propose a training-free strategy to enhance offsite-tuning by leveraging the similarities among neural networks.

**Compression methods for LLMs** To address resource-constrained scenarios, various technologies such as knowledge distillation (Hinton et al., 2015; Sun et al., 2019; Gordon et al., 2020), pruning (Chen et al., 2020; Liu et al., 2022), and quantization (Tao et al., 2022) have been employed in pre-trained models such as BERT, T5, and GPT-2. However, with the increasing size of LLMs, such as GPT-3 which consists of 96 layers, knowledge distillation becomes challenging. With the increasing popularity of post-quantization, methods such as GPTQ (Frantar et al., 2023) and QLoRA (Dettmers et al., 2023) have emerged to reduce the computational resources required by LLMs. However, these methods still encounter challenges related to loss of precision and model leakage, posing threats to the privacy and safety of LLMs. SparseGPT (Frantar and Alistarh, 2023) demonstrated, for the first time, the possibility of pruning large-scale generative pretrained transformer (GPT) family models to achieve at least 50% sparsity in a single opera-

tion. In comparison to the aforementioned methods, LayerDrop is an effective technique for reducing the parameters of LLMs while maintaining comparable performance. Fan et al. (2020) explored the use of LayerDrop to reduce the depth of transformers, but this approach relies on a specialized training strategy. Building on this, Zhang and He (2020) proposed the concept of layer drop to expedite training. These studies primarily focus on training with layer drop during the training process rather than during post-training. Sajjad et al. (2023) investigated various strategies for implementing layer drop on pre-trained models such as BERT, XLNet, and GPT-2, with a particular emphasis on the application of heuristic rules. Building upon the concept of "lottery tickets" (Frankle and Carbin, 2019), which involves discovering sparse subnetworks within trained dense networks that can achieve comparable accuracy, Jordao et al. (2023) confirmed the presence of winning tickets when layers are pruned in vision models. Insufficient exploration has been conducted on the application of layer dropping in large-depth neural networks such as LLMs.

**Neural network similarity** primarily encompasses representation similarity and functional similarity. Representation similarity refers to the similarity of hidden states across all layers, while functional similarity indicates that two models exhibit the same behavior on the given inputs (Klabunde et al., 2023). By measuring the similarity of neural network representations, we can gain insights into the reasons behind variations in model behavior. Kornblith et al. (2019) revisited previous methods for comparing neural network representations and introduced centered kernel alignment (CKA), a technique that measures the relationship between representations without being affected by high dimensionality or the number of data points. Extensive exploration has been conducted in the field of computer vision. In comparison to CNN models, Raghu et al. (2021) discovered significant differences in the representation structure between Vision Transformers (ViTs) and convolutional networks. ViTs exhibit highly similar representations across all layers, whereas ResNet models demonstrate lower similarity between lower and higher layers, indicating the absence of a block structure in the representation of transformer models. In the study of large language models, researchers have analyzed representation similarity across different pre-trained models, comparing them within various model families (Wu et al., 2020), and have investigated the dynamics of embeddings during fine-tuning (Merchant et al., 2020). Phang et al. (2021) discovered the clustering of layer representations in fine-tuned transformers, providing evidence for the presence of layer redundancy in LLMs (Dalvi et al., 2020). As the size increases, we observe that clustering also occurs in original pre-trained models, not just in specialized models. Recognizing the wide applicability of this concept in various fields, recent studies have aimed to gain insights into the functional aspects of LLMs. nostalgebraist (2020) directly mapped the hidden states of intermediate layers to the final classification layer to obtain hidden predictions. Considering the phenomenon of representation drift within hidden layers, especially in lower layers, Belrose et al. (2023) propose tuned-lens, a method that trains projections for each layer using a small amount of pre-trained corpus. These methods indicate that the functional behaviors of hidden layers also undergo subtle changes as the depth of the layers increases. Building upon these observations, Merullo et al. (2023) discovered that language models implement simple word2vec-style vector arithmetic (Mikolov et al., 2013), which sheds light on the prediction process of LLMs. Instead of using the transformation layer for decoding, Bills et al. (2023) attempted to interpret GPT-2 hidden states using GPT-4 (OpenAI, 2023), opening up the possibility of automatically aligning the behavior of models with AI.

| OpenBookQA |
| --- |
| Which organism cannot specialize? 
 *protozoa* |
| A person can grow cabbage in January with the help of what product? 
 *Green house* |

| PIQA |
| --- |
| Question: To fight Ivan Drago in Rocky for sega master system. 
 Answer: 
 *You have to defeat Apollo Creed and Clubber Lang first.* |
| Question: Make outdoor pillow. 
 Answer: 
 *Blow into trash bag and tie with rubber band.* |

| SciQ |
| --- |
| A wetland is an area that is wet for all or part of the year. Wetlands are home to certain types of plants. Question: What is an area of land called that is wet for all or part of the year? 
 Answer: 
 *wetland* |
| Question: Surface waters are heated by the radiation from? 
 Answer: 
 *the sun* |

| RACE |
| --- |
| Article: I am a psychologist. I first met Timothy, a quiet, overweight eleven-year-old boy, when his mother brought him to me to discuss his declining grades. A few minutes with Timothy were enough to confirm that his self-esteem and general happiness were falling right along with _ . I asked about Timothy's typical day. He awoke every morning at six thirty so he could reach his school by eight and arrived home around four thirty each afternoon. He then had a quick snack, followed by either a piano lesson or a lesson with his math tutor. He finished dinner at 7 pm, and then he sat down to do homework for two to three hours. Quickly doing the math in my head, I found that Timothy spent an average of thirteen hours a day at a writing desk. 
 What if Timothy spent thirteen hours a day at a sewing machine instead of a desk? We would immediately be shocked, because that would be called children being horribly mistreated. Timothy was far from being mistreated, but the mountain of homework he faced daily resulted in a similar consequence –he was being robbed of his childhood. In fact, Timothy had no time to do anything he truly enjoyed, such as playing video games, watching movies, or playing board games with his friends. Play, however, is a crucial part of healthy child development. It affects children's creativity, their social skills, and even their brain development. The absence of play, physical exercise, and freefrom social interaction takes a serious toll on many children. It can also cause significant health problems like childhood obesity, sleep problems and depression. 
 Experts in the field recommend the minutes children spend on their homework should be no more than ten times the number of their grade level. As a fifthgrader, Timothy should have no more than fifty minutes a day of homework (instead of three times that amount). Having an extra two hours an evening to play, relax, or see a friend would soundly benefit any child's life quality. 

 Question: According to the passage, how long should a thirdgrader spend a day doing homework? 

 Answer: 
 *No more than thirty minutes.* |

| BoolQ <support set> |
| --- |
| Phantom pain – Phantom pain sensations are described as perceptions that an individual experiences relating to a limb or an organ that is not physically part of the body. Limb loss is a result of either removal by amputation or congenital limb deficiency. However, phantom limb sensations can also occur following nerve avulsion or spinal cord injury. 
 Question: is pain experienced in a missing body part or paralyzed area? 
 Answer: 
 *yes* |

Table 5: Instructions format of Multi-Choice QA task.

| ARC-E |
|---|
| Question: Which technology was developed most recently?
Answer:
*cellular telephone* |
| Question: A student hypothesizes that algae are producers. Which question will best help the student determine if this is correct?
Answer:
*Do algae use sunlight to make food?* |

| ARC-C |
|---|
| Question: Juan and LaKeisha roll a few objects down a ramp. They want to see which object rolls the farthest. What should they do so they can repeat their investigation?
Answer:
*Record the details of the investigation.* |
| Question: High-pressure systems stop air from rising into the colder regions of the atmosphere where water can condense. What will most likely result if a high-pressure system remains in an area for a long period of time?
Answer:
*drought* |

| WebQS |
|---|
| Question: what is the name of justin bieber brother?
Answer:
*Jazmyn Bieber* |
| Question: what character did natalie portman play in star wars?
Answer:
*Padmé Amidala* |

| TriviaQA <support set> |
|---|
| Question: What star sign is Jamie Lee Curtis?
Answer:
*Scorpio* |
| Question: Which Lloyd Webber musical premiered in the US on 10th December 1993?
Answer:
*Sunset Boulevard* |

| HellaSwag |
|---|
| Roof shingle removal: A man is sitting on a roof. He
*starts pulling up roofing on a roof.* |
| Clean and jerk: A lady walks to a barbell. She bends down and grabs the pole. The lady
*stands and lifts the weight over her head.* |

| CoPA <support set> |
|---|
| The man turned on the faucet therefore
*water flowed from the spout.* |
| The girl found a bug in her cereal therefore
*she lost her appetite.* |

Table 6: Instructions format of Closed-Book QA and Sentece Completion task.

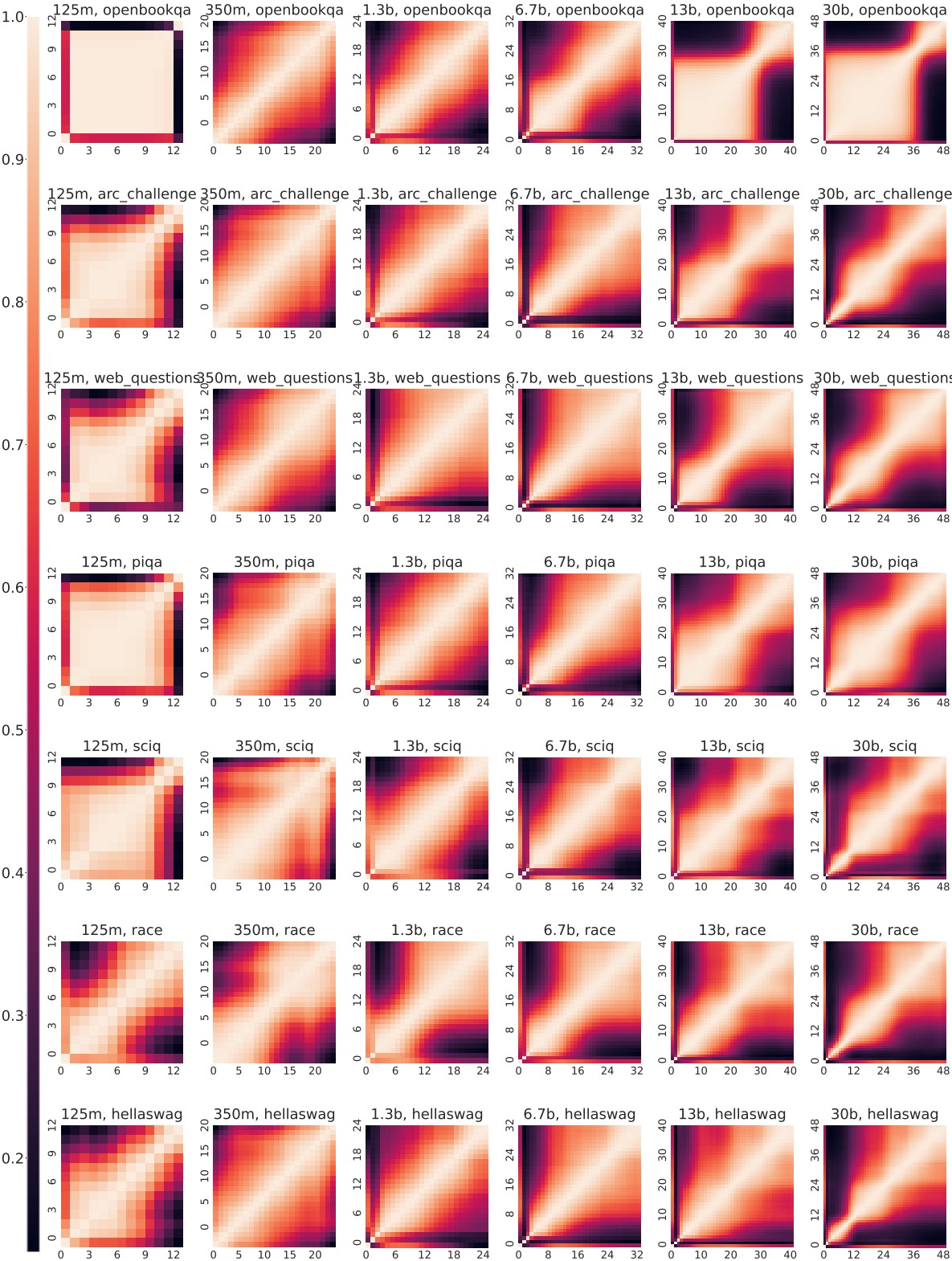

Figure 11: The table displays the representation similarity among layers of OPT models (Zhang et al., 2022) with different sizes, across five datasets with weighted mean pooling strategy. The rows in the table represent the layer similarity of each dataset across the 125M, 1.3B, 6.7B, and 13B OPT models. Notably, the characteristics of the Wikitext dataset differ from the rest because it is a language modeling task similar to the pre-training objective. (Phang et al., 2021) introduced that fine-tuned transformers exhibit clusters of similar representations across layers, which explains the emerging block structure of the Wikitext dataset even in small language models. In contrast, the remaining four datasets are associated with question answering tasks, which pose challenges for smaller models. Consequently, the representation of these tasks demonstrates abnormal similarity in the intermediate layers of the 125M OPT model, suggesting a relatively lower level of comprehension in it.

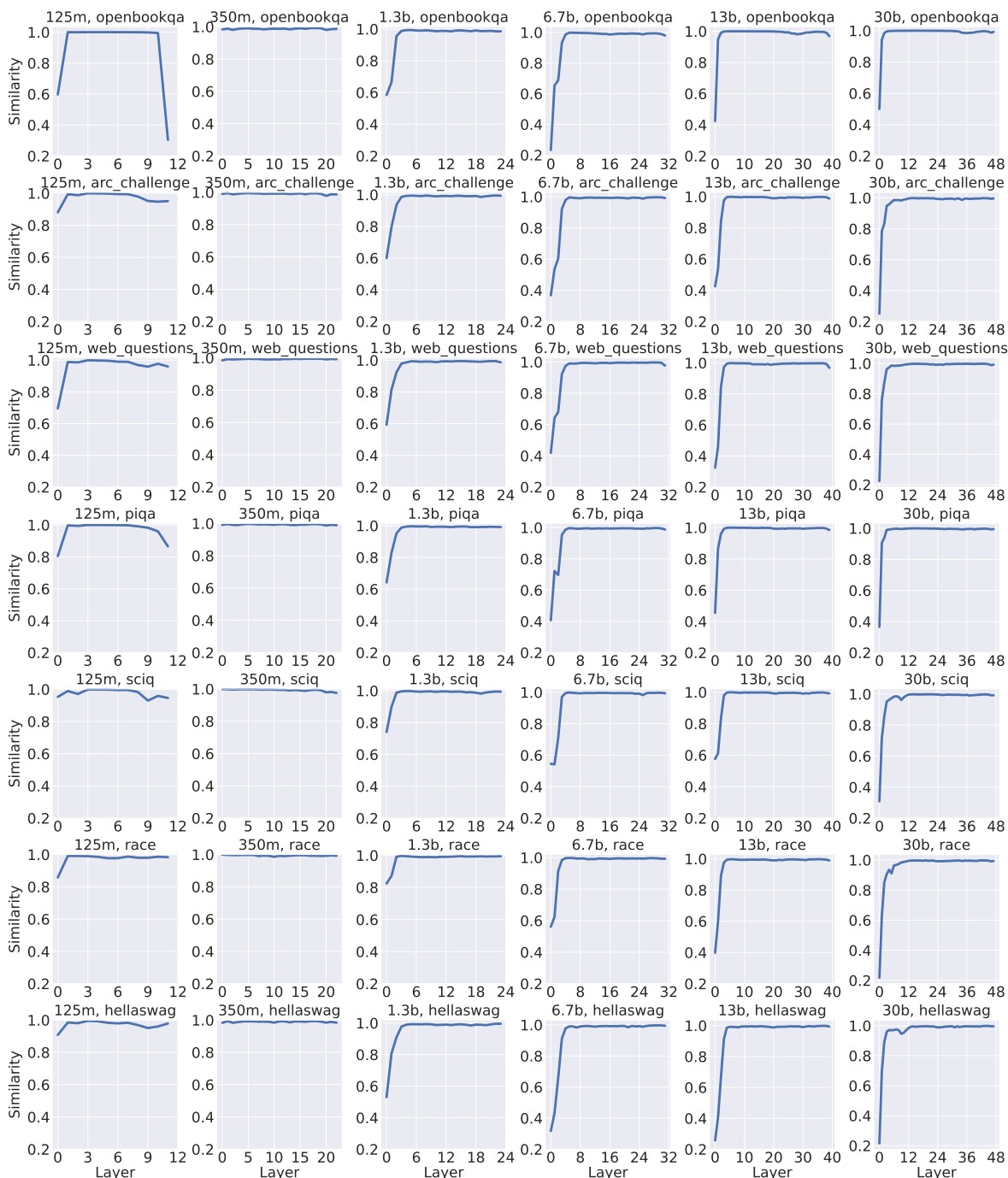

Figure 12: The table displays the representation similarity among layers of OPT models (Zhang et al., 2022) with different sizes, across five datasets with weighted mean pooling strategy.

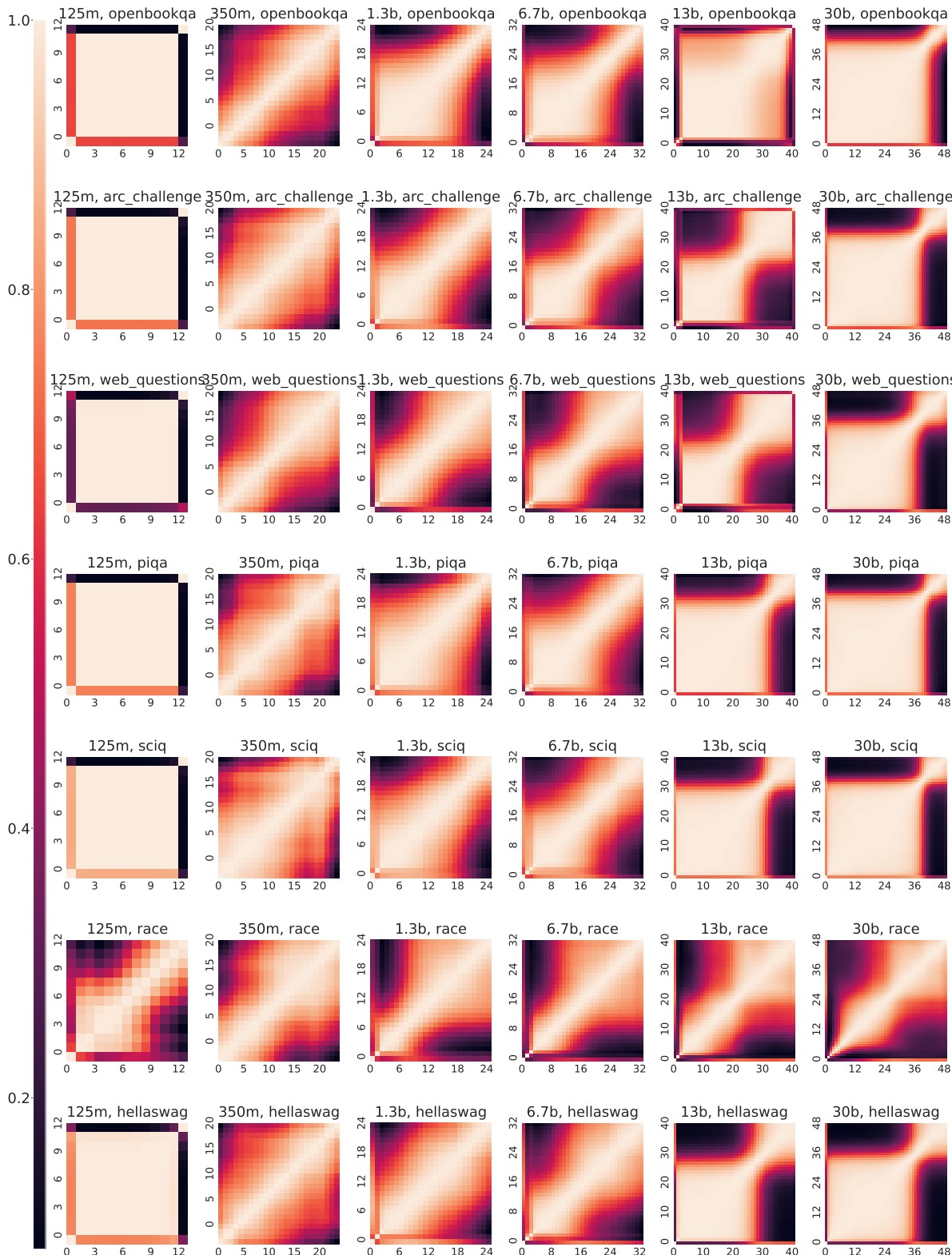

Figure 13: The table displays the representation similarity among layers of OPT models (Zhang et al., 2022) with different sizes, across five datasets with simple mean pooling strategy.

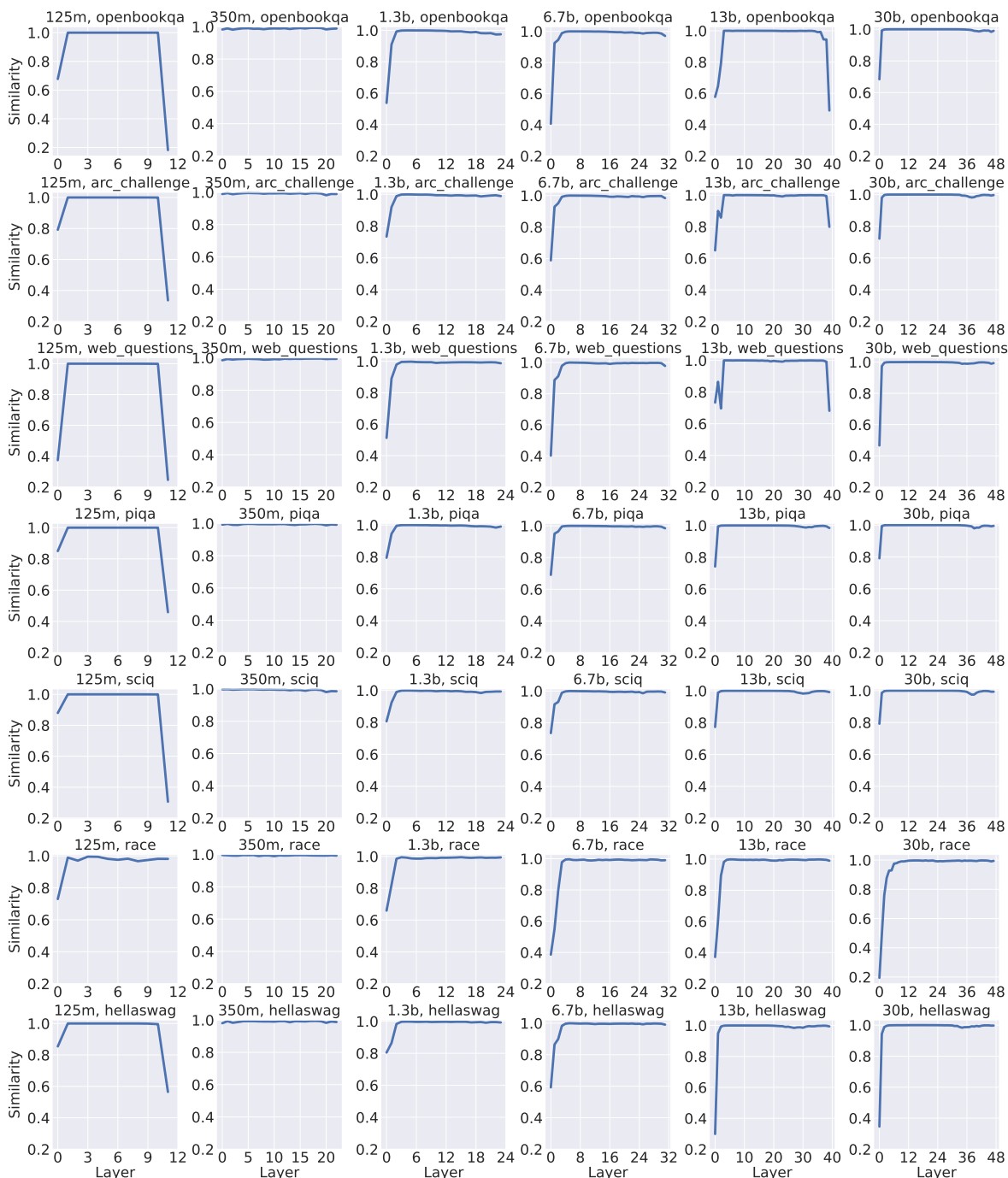

Figure 14: The table displays the representation similarity among layers of OPT models (Zhang et al., 2022) with different sizes, across five datasets with simple mean pooling strategy.

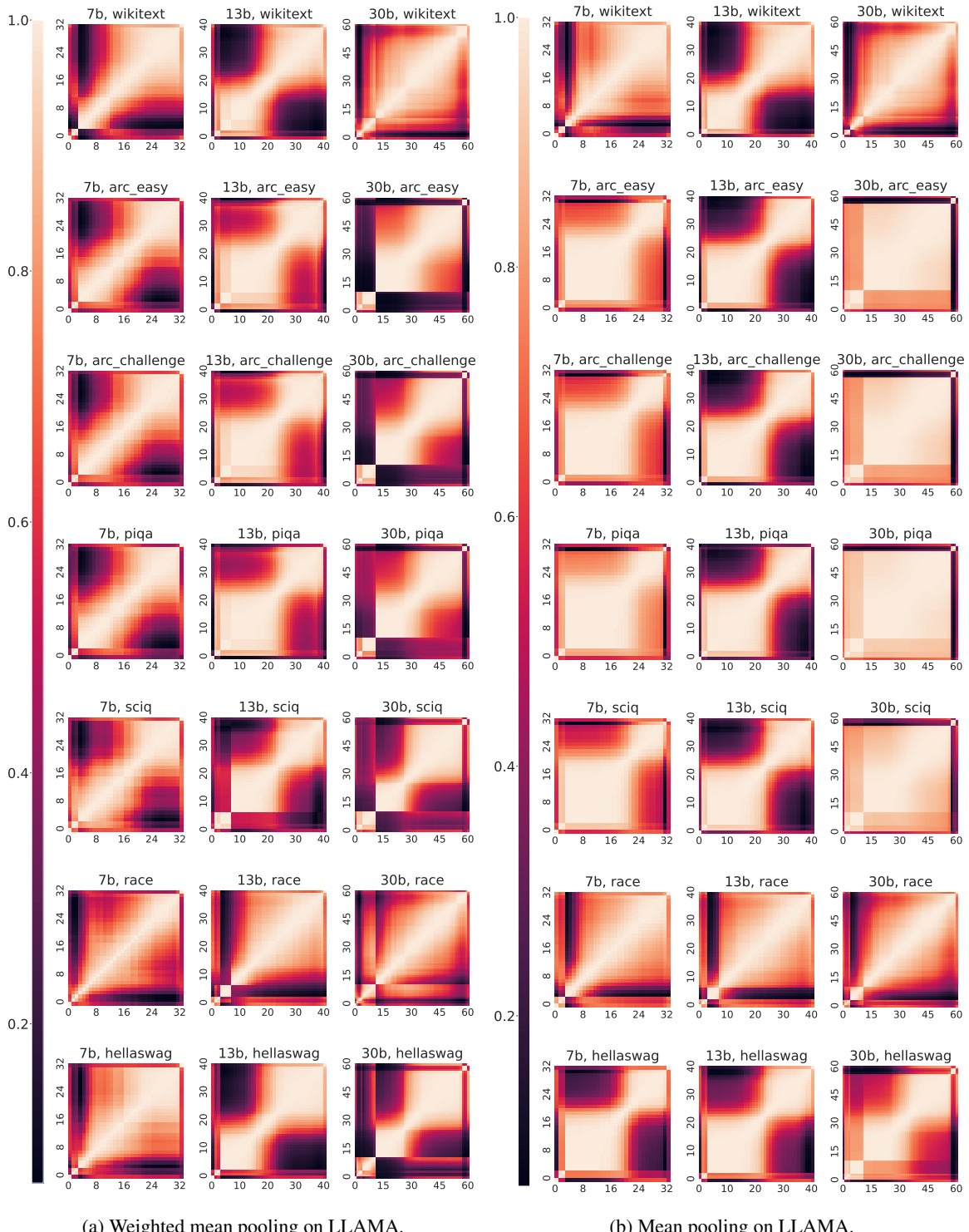

(a) Weighted mean pooling on LLAMA.

(b) Mean pooling on LLAMA.

Figure 15: This table illustrates the presence of a *modular structure* in LLAMA models. We observe that weighted mean pooling performs well on LLAMA models, while direct mean pooling may lead to information loss and abnormal structures, particularly in the case of 30B models.