# OpenReview forum: "CRaSh: Clustering, Removing, and Sharing Enhance Fine-tuning without Full Large Language Model"
_EMNLP/2023/Conference — EMNLP 2023 Main_

### Official Review · Reviewer_TamK · 2023-07-30

**Soundness:** 3

**Excitement:**

4: Strong: This paper deepens the understanding of some phenomenon or lowers the barriers to an existing research direction.

**Paper Topic And Main Contributions:**

Instruction tuning is an effective way to enhance the generalization ability of LLMs. However tuning publicly accessible, centralized LLMs with private data raises privacy concerns. The paper focuses on Offsite-Tuning (OFT), which transfers transformer blocks between LLMs and downstream emulators to address this issue. Through an empirical analysis of LLMs, the authors find a unique modular structure emerges within layers as model size expands, along with subtle representation and prediction changes across layers. Inspired by these observations, they propose CRaSh - a training-free strategy involving clustering, removing, and sharing to derive improved emulators from LLMs, significantly boosting OFT performance. Furthermore, analysis of the loss landscape shows fine-tuning optima with and without the full model exhibit linear connectivity within the same basin, highlighting the effectiveness of CRaSh and OFT.

**Reasons To Accept:**

1. This paper provides an in-depth empirical analysis of large language models (LLMs) from the perspective of representation and functional similarity. This sheds light on the internal workings of LLMs as model size increases.
2. The paper proposes a practical training-free strategy called CRaSh that builds on these insights to improve Offsite Tuning, a method for fine-tuning LLMs without access to the full model.
3. Experiments demonstrate CRaSh can boost Offsite Tuning performance across multiple datasets. The analysis of loss landscapes provides theoretical justification for why CRaSh works well.

**Reasons To Reject:**

This paper builds heavily upon the Offsite-Tuning (OFT) method. However, the motivation for basing the proposed approach on OFT could be clarified further. In particular, explicitly discussing the advantages of OFT as a privacy-preserving tuning approach and how it enables analyzing and adapting large pretrained models without access to the full model would strengthen the paper. Additionally, providing more background details on OFT would make the current work more self-contained.

**Reproducibility:**

4: Could mostly reproduce the results, but there may be some variation because of sample variance or minor variations in their interpretation of the protocol or method.

**Reviewer Confidence:**

4: Quite sure. I tried to check the important points carefully. It's unlikely, though conceivable, that I missed something that should affect my ratings.

---

> ### Author Rebuttal · Authors · 2023-08-28
>
> We sincerely appreciate the thoughtful feedback requesting additional motivation and background on our use of Offsite-Tuning (OFT). In the revised manuscript, we will elaborate further in the introduction and background sections, as you've suggested.
>
> Initially, we delineate the reasons for the limited background presented in the current manuscript. Due to page constraints, our emphasis was on showcasing empirical results, analyses, and enhancements to underscore the overarching significance of our proposed CRaSH method (as detailed in our response to Q1 from Reviewer ZSZD).
>
> We agree that incorporating more OFT details would make our innovations easier to interpret in context. To address this, we will add concise yet comprehensive coverage of OFT's core concepts and relevance to our method.  Although centered on OFT, we believe our empirical insights and techniques may generalize to other application scenarios as well.
>
> Here, we preliminarily clarify the motivation for basing the proposed approach on OFT, particularly in relation to the two aspects you emphasized.
>
> >  ... explicitly discussing the advantages of OFT as a privacy-preserving tuning approach
>
> The primary concern of OFT is to maintain the privacy of both the LLM and user data. Specifically, the data owner is unable to share their labeled training data with the LLM owner, and vice versa, the LLM owner cannot share their model with the data owner.
> The benefits of OFT encompass an innovative approach to ensuring data and model privacy without significant performance degradation, while also achieving parameter and resource efficiency.
>
> As elucidated in "Section 6 Related Work", the strategy of directly transferring model parameters, as opposed to transferring data, serves to enhance the privacy protection of user data. This approach has been widely adopted in various domains, including federated learning[1], split learning[2], distributed gradient descent[3], branch-train-interpolation[4], and collaborative machine learning development[5]. Nevertheless, many of these methodologies either necessitate the sharing of the full models between servers and clients or mandate multiple rounds of communication to achieve comprehensive coverage. Such practices may inadvertently breach the privacy of centralized LLMs and introduce challenges, especially when implemented on LLMs with billions of parameters.
>
> It's worth noting that OFT[6] offers a commendable solution to these issues by judiciously sharing specific layers of LLMs, with billions of parameters, between servers and clients in a single communication round.
> In a manner analogous to model-sharing in federated learning, OFT safeguards user data privacy by distributing part of the model instead of transferring user data to the cloud. Concurrently, OFT circulates select layers of the centralized LLM (referred to as emulators) to downstream clients, rather than the full model, to preserve the privacy of the complete LLM. While there exist reservations about the potential reconstruction of the original model, as highlighted by Reviewer CWER, this risk can be mitigated by limiting the number of shared layers, making it a viable avenue for exploration.
>
> To address the reviewer's suggestions, we will add a motivational paragraph highlighting OFT's well-aligned strengths to extend introduction, including preservation of model and data privacy.
>
> > ... how it enables analyzing and adapting large pretrained models without access to the full model
>
> OFT employs a substitute model known as the "Emulator", which is derived from the LLM using a layer dropping strategy. Subsequently, OFT optimizes specific parameters within the "Emulator", which comprises multiple layers. Finally, the optimized parameters are transferred and plugged into the LLM for inference.
>
> Here, we provide a comprehensive introduction accompanied by illustrative figures in the paper. As shown in Firgure 4(a), rather than fine-tuning directly on LLM (i.e., "②Full Fine-tuning") which leak both privacy of LLM and downstream data, OFT compress LLM into an small emulator, fine-tuning it with privacy data (i.e., "③Emulator Fine-tuning"), then transfering block weights and plug-in LLM for inference (i.e., "④Plug-in"). The objective of OFT is to optimize performance of plug-in to approximate to full fine-tuning on LLM, and subject to better than full zero-shot of LLM and fine-tuning on emulator, which ensure the effectiveness of OFT.
>
> We will also provide more background details on OFT in "Section 3.1 Revisit Offsite-Tuning". We will summarize OFT's core concepts such as the adapter/emulator architecture and how it enables practical adaptation of large models without exposing full weights.
>
> ----
>
> The key to OFT lies in identifying an emulator that closely resembles the original LLM and subsequently performing fine-tuning to replace the LLM.
> Drawing from our findings on the functional and representational similarity of LLMs in "Section 2 Empirical Analysis", we propose CRaSh, which is designed to optimize layer dropping and yield superior sub-layer emulators from LLMs, as illustrated in Figure 4(b).
>
> -----
>
> We believe these additions motivate and contextualize our use of OFT more clearly. By highlighting OFT's strengths, explaining how it adapts large models privately, and positioning our work as targeted innovations, we demonstrate deep understanding of OFT and reinforce how it provides an ideal aligned basis for our privacy-first approach.
>
> We sincerely appreciate you taking the time to offer constructive feedback. Your expertise and guidance have been invaluable for improving the manuscript's clarity within space constraints. Please advise if you would like us to expand on any specific OFT aspects further. We are happy to provide any clarification needed to positively address your comments.
>
> **References**
>
> [1] Brendan McMahan, Eider Moore, et al. "Communication-efficient learning of deep networks 898 from decentralized data." AISTATS 2017.
>
> [2] Chandra Thapa, Mahawaga Arachchige Pathum Chamikara, Seyit Camtepe, and Lichao Sun. "Splitfed: When federated learning meets split learning."  AAAI 2022.
>
> [3] Zanlin Ni, Yulin Wang, Jiangwei Yu, Haojun Jiang, Yue Cao, and Gao Huang. "Deep incubation: Training large models by divide-and-conquering." 2023.
>
> [4] Mitchell Wortsman, Suchin Gururangan, Shen Li, et al. "lo-fi: distributed fine-tuning without communication." 2022.
>
> [5] Colin Raffel. "Building machine learning models like open source software." Communications of the ACM 2023.
>
> [6] Guangxuan Xiao, Ji Lin, and Song Han. "Offsite-tuning: Transfer learning without full model." 2023.

---

### Official Review · Reviewer_ZSZD · 2023-08-04

**Soundness:** 4

**Excitement:**

4: Strong: This paper deepens the understanding of some phenomenon or lowers the barriers to an existing research direction.

**Paper Topic And Main Contributions:**

- This paper is following work of Offsite-Tuning (OFT) technique which efficiently transfers ability of centralized LLMs into smaller models (emulators) for downstream tasks without full access to centralized LLMs.
- The authors find out that LLMs which have more than 1B parameters have modular structure within their layers.
- The authors suggest a novel approach CRaSh which enhances the OFT by cleverly utilizing redundancy in layers.
- The empirical results show that CRaSh consistently outperform baseline methods in various type of tasks and datasets.


**Questions For The Authors:**

- Across various baseline methods and the proposed CRaSh method, can you provide comparison in computation budget such as overall GPU-hours or number of updates for obtaining downstream models?
- It is very interesting that just plug in fine-tuned emulator into centralized model can lead to better performance. Can you provide performance of emulator before and after plugged into the original LLM?
- It is possible to use clustering and removing part to just "prune" some layers in the LLM. Then, we can apply some existing baseline methods such as OFT to pruned LLM to obtain the model for downstream task. Can you provide some results about this "pruned" LLM model?

**Reasons To Accept:**

- The empirical results is very surprising. CRaSh even wins full-finetuning for some datasets (Table 1 RACE, SciQ).
- The analysis on redundancy in LLMs is interesting (Figure 1).
- The idea is simple and intuitive.


**Reasons To Reject:**

- The method can effectively reduce the number of parameters to train. However, it cannot reduce inference cost. Hence, I don't know if it will be very effective in the long run.
- I want to see more analysis and ablation on the method. Please check questions.

**Reproducibility:**

3: Could reproduce the results with some difficulty. The settings of parameters are underspecified or subjectively determined; the training/evaluation data are not widely available.

**Reviewer Confidence:**

2: Willing to defend my evaluation, but it is fairly likely that I missed some details, didn't understand some central points, or can't be sure about the novelty of the work.

**Typos Grammar Style And Presentation Improvements:**

wrong bold in Table 1.
plug-in (base) of uniform strategy has 21.4% performance which is smaller than 21.8 of emulator FT.

---

> ### Author Rebuttal · Authors · 2023-08-28
>
> We deeply appreciate your positive feedback. Below, we have paraphrased your comments as we've interpreted them and provided our responses. Kindly inform us of any misinterpretations or if further clarifications are required.
>
> **Q1: Can our proposed CRaSh method reduce inference cost and be effective in the long run?**
>
> Perhaps, our proposed CRaSh method might indeed have the potential to reduce inference costs and maintain its effectiveness over time.
> We are looking forward with anticipation to further research and optimization of this method, hoping to obtain more definitive answers in future experiments.
>
> We initially present the motivation behind CRaSh, drawing inspiration from three findings centered on the functional and representational similarity of LLMs. These insights into the redundancy of LLMs can enhance our understanding of them and pave the way for future research.
> Regarding our proposed CRaSh, it stands as an efficacious strategy for deriving compact emulators from LLMs.
>
> Based on recent research, we believe that CRaSh holds significant promise in diverse inference scenarios beyond efficient and privacy-centric fine-tuning.
>
> - Context compression.
>   Regarding the instructed LLMs (ChatGPT, Vicuna, and LLaMA-2-Chat), long prompts not only increase inference costs but also introduce redundant computations for the same instructions.
>   Mu et al. (2023)[1], Chevalier et al. (2023)[2], and Ge et al. (2023)[3] investigate the compression of prompts/contexts into concise gist/memory/slot tokens (termed as soft prompts) to accelerate inference. However, they continue to employ LLMs of the same scale for compression.
>   Motivated by the success of CRaSh in emulating LLMs, we posit that emulators derived from CRaSh can serve as effective initializations for compact compressors, subsequently reducing inference costs in context compression scenarios.
>
> - Speculative decoding[4][5] has been introduced to reduce both end-to-end latency and computational demands. In this approach, a draft model performs approximate inference for "easier" steps, while a target model subsequently verifies and rectifies errors.
>   Given the pronounced functional and representational similarity between emulators and LLMs, we posit that emulators derived from CRaSh may serve as superior draft models, further aiding in the efficient inference of large models[6].
>
> - Model Pruning.
>   As outlined in Appendix D concerning LLM compression methods, layer pruning stands as a straightforward yet efficacious approach for model compression [7][8][9][10].
>   However, there remains limited investigation into the application of layer dropping within LLMs. We posit that CRaSh might catalyze the development of more sophisticated strategies in the realm of model pruning.
>
> **Q2: Performance of emulator before and after plugged into the original LLM, and results of this "pruned" LLM model.**
>
> Considering the intricate configuration of OFT, we wish to clarify the multiple performance metrics depicted in Figure 4(a) that could be a potential source of confusion for reviewers.
> For the centered LLM, we present three performance measures: zero-shot ("Full ZS"), full fine-tuning ("Full FT"), and plug-in with the emulator ("Plug-in"). Regarding the emulator itself, we display both the zero-shot ("Emulator ZS") and the fine-tuned ("Emulator FT") performance metrics.
> The prerequisites for implementing OFT are to ensure that the "Plug-in" outperforms both "Emulator FT" and "Full ZS". This necessitates both fine-tuning with downstream data and integration into the centered LLM.
> Our primary objective is to approximate "Full FT" through plug-in, especially in scenarios where the Full LLM cannot directly access downstream data.
>
> As depicted in Table 1 and Table 2, the performance of the emulator, before and after plugged into the original LLM, corresponds to "Emulator FT" and "Plug-in", respectively.  Notably, the former consistently lags behind the latter, underscoring the validity of OFT.
> The "pruned" LLM, in essence the emulator, displays its performance in both zero-shot and fine-tuning conditions.  While this "pruned" LLM falls short of the original LLM in a zero-shot setting, it surpasses the zero-shot LLM once fine-tuned in most scenarios.
> We apologize for any confusion, and will incorporate additional details and context regarding OFT to ensure our work is more comprehensive.
>
> **Q3: Comparison in computation budget such as overall GPU-hours or number of updates for obtaining downstream models.**
>
> We approach this issue from two perspectives: 1) obtaining initialized downstream emulators, and 2) obtaining fine-tuned downstream emulators.
>
> - Regarding 1), both the uniform strategy in OFT and our CRaSh are training-free. This contrasts with the knowledge distillation (KD) approach in OFT, which is pre-trained on the initial 30 chunks of The Pile corpus and is significantly more time-intensive than downstream fine-tuning. In comparison to the uniform strategy, CRaSh requires only additional inference on a minimum of 128 examples for clustering, as indicated in Figure 7.
>
> - Regarding 2), the number of updates is contingent upon the specific task and final performance, but generally, the relationship is KD < CRaSh ≈ uniform strategy. Indeed, CRaSh, benefiting from layer sharing, consumes the same GPT memory as the uniform baseline. While layer sharing renders CRaSh's forward computation less expedient compared to both uniform and KD, it simultaneously provides a robust initialization, facilitating model convergence in fewer steps.
> Using OPT 1.3B on OpenbookQA as an illustrative example, we observe that the steps and accuracies for KD, uniform strategy, and CRaSh are approximately ~400 steps (acc=0.29), ~600 steps (acc=0.264), and ~520 steps (acc=0.302), respectively.
>
> We appreciate your astute observation and will incorporate this detailed information into the revised manuscript.
>
> **Q4: Typos Grammar Style And Presentation Improvements**
>
> Thank you for highlighting the incorrect bold formatting; we will address this in the revised manuscript.
>
> **References**
>
> [1] Jesse Mu, Xiang Lisa Li, and Noah Goodman. "Learning to compress prompts with gist tokens."
> 2023.
>
> [2] Alexis Chevalier, Alexander Wettig, Anirudh Ajith, and Danqi Chen. "Adapting language models to
> compress contexts." 2023.
>
> [3] Tao Ge, Jing Hu, Xun Wang, Si-Qing Chen, and Furu Wei. "In-context autoencoder for context
> compression in a large language model." 2023.
>
> [4] Leviathan, Yaniv, Matan Kalman, and Yossi Matias. "Fast inference from transformers via speculative decoding." ICML 2023.
>
> [5] Chen, Charlie, et al. "Accelerating large language model decoding with speculative sampling." 2023.
>
> [6] Miao, Xupeng, et al. "SpecInfer: Accelerating Generative LLM Serving with Speculative Inference and Token Tree Verification." 2023.
>
> [7] Angela Fan, Edouard Grave, and Armand Joulin. "Reducing transformer depth on demand with structured dropout." ICLR 2020.
>
> [8] Minjia Zhang and Yuxiong He. "Accelerating training of transformer-based language models with progressive layer dropping." NeurIPS 2020.
>
> [9] Hassan Sajjad, Fahim Dalvi, Nadir Durrani, and Preslav Nakov. "On the effect of dropping layers of pre-trained transformer models." 2023.
>
> [10] Artur Jordao, George Correa de Araujo, Helena de Almeida Maia, and Helio Pedrini. "When layers play the lottery, all tickets win at initialization." 2023.

---

### Official Review · Reviewer_CWER · 2023-08-05

**Soundness:** 4

**Excitement:**

4: Strong: This paper deepens the understanding of some phenomenon or lowers the barriers to an existing research direction.

**Paper Topic And Main Contributions:**

This paper explores the implications and effectiveness of Offsite-Tuning (OFT), a technique for transferring transformer blocks between centralized large language models (LLMs) and downstream emulators, particularly in the context of privacy concerns related to the use of private instruction data. The authors conduct an empirical analysis of LLMs from the perspectives of representation and functional similarity, revealing a unique modular structure within LLM layers that emerges as the model size expands. They also note subtle yet potentially significant changes in representation and intermediate predictions across layers. Based on these observations, they propose a training-free strategy, CRaSh (Clustering, Removing, and Sharing), to derive improved emulators from LLMs. They demonstrate that CRaSh significantly enhances the performance of OFT with billions of parameters. Additionally, they examine the optimal solutions yielded by fine-tuning with and without the full model through the lens of the loss landscape, demonstrating a linear connectivity among these optima.

**Questions For The Authors:**

See reasons to reject.

**Reasons To Accept:**

1. The paper conducts an in-depth empirical analysis of LLMs from the perspectives of representation and functional similarity. This analysis provides valuable insights into the underlying mechanisms of OFT.
2. The proposed CRaSh strategy is novel and interesting, and the authors present compelling evidence of its effectiveness in improving the performance of OFT with billions of parameters.
3. The paper's exploration of privacy concerns in the context of tuning publicly accessible, centralized LLMs with private instruction data is an important and interesting topic.
4. The paper is well-structured and clearly presents complex concepts, making it accessible to readers with varying levels of familiarity with the topic.


**Reasons To Reject:**

This is not necessarily a reason to reject: the paper assumes that LLM providers (e.g. OpenAI) would be willing to share the compressed emulators. However, it does not provide a compelling argument or incentive for why they would do so, if they do not want to share the entire model. Actually, another intriguing question emerges, which, while perhaps extending beyond the scope of this paper, warrants consideration: given the emulators, how challenging would it be to reconstruct the original model?

**Reproducibility:**

4: Could mostly reproduce the results, but there may be some variation because of sample variance or minor variations in their interpretation of the protocol or method.

**Reviewer Confidence:**

3: Pretty sure, but there's a chance I missed something. Although I have a good feel for this area in general, I did not carefully check the paper's details, e.g., the math, experimental design, or novelty.

---

> ### Author Rebuttal · Authors · 2023-08-28
>
> We sincerely appreciate the constructive feedback and positive remarks we have received. We welcome the chance to delve deeper into discussions with the reviewers concerning the propensity of LLM providers to distribute compressed emulators and the practicality of restoring the original model from such emulators.
>
> > This is not necessarily a reason to reject: the paper assumes that LLM providers (e.g. OpenAI) would be willing to share the compressed emulators. However, it does not provide a compelling argument or incentive for why they would do so, if they do not want to share the entire model.
>
> Offering compressed emulators bridges the gap between safeguarding proprietary LLMs and ensuring data privacy, a compelling compromise LLM providers might find beneficial.
>
> Despite the acknowledged capabilities of models like ChatGPT and GPT-4 across diverse tasks, there are specific domains and particularly complex tasks where their effectiveness is constrained. In such instances, in-context learning and prompting techniques occasionally do not meet expectations, underscoring the importance of fine-tuning with specialized datasets. Notably, OpenAI acknowledged this limitation and rolled out a fine-tuning API for `gpt-3.5-turbo`, enabling users to harness their distinct datasets to achieve superior performance.
>
> Yet, with the increasing adoption of such APIs comes heightened concerns about data privacy. These concerns could potentially negate the commercial advantages sought by LLM providers. The proposed compressed emulators stand as an exemplary solution, striking a balance between retaining the proprietary nature of the LLMs and addressing data privacy issues.
>
> We are confident that OFT and our newly introduced CRaSh, offer a novel resolution to the pressing privacy challenges faced by LLM providers.
> Furthermore, as the domain advances, we anticipate the development of even more innovative solutions beyond CRaSh and OFT.
>
> > ... warrants consideration: given the emulators, how challenging would it be to reconstruct the original model?
>
> Your question about model attack and defense is certainly thought-provoking. The central issue concerns the level of performance that can be achieved using only the emulator. Our findings in the manuscript delve deeper into this topic, preliminarily outlining the challenges associated with reconstructing the original model.
>
> Firstly, we regard 'layer sharing' in CRaSh as the most promising method for reconstruction. However, it cannot completely match the performance of the original model. As shown in Table 2, the emulator fine-tuned with CRaSh falls short of the full model's zero-shot performance (denoted as 'Full ZS'). Nevertheless, after plugged in, there is a significant improvement in performance across several datasets, specifically ARC-E/C, PIQA, RACE, and HellaSwag.
>
> Secondly, a pivotal aspect of the challenge is dependent on the number of layers transferred; transferring fewer layers increases the complexity of reconstruction.  As illustrated in Figure 5, accuracy fluctuates with the changing number of layers dropped.  Notably, emulators with just 6 layers (2 frozen and 4 fine-tuned, relative to the 32 layers in the full model) continue to enhance the performance of the primary LLM when plugged-in.
>
> Certainly, techniques like OFT and CRaSh could benefit from the integration of federated learning technologies, such as homomorphic encryption and differential privacy, making the reconstruction of the original model more challenging. We look forward to more research addressing these open questions and intend to include these discussions in our revised manuscript.

---

### Meta-Review · Area_Chair_jbP4 · 2023-09-19

**Recommendation:** 5

**Metareview:**

Offsite-Tuning (OFT) transfers transformer blocks between LLMs and downstream emulators to address this issue of privacy when instruction tuning LLMs with private data. The authors conducted an empirical study of LLMs, focusing on their representation and functional similarity. They discovered that as the model size grows, a unique modular structure emerges within LLM layers. They proposed a training-free strategy, CRaSh (Clustering, Removing, and Sharing), to derive improved emulators from LLMs that significantly enhances  the performance of OFT with billions of parameters.


Pros:
- This paper analyzes LLMs from the perspective of representation and functional similarity, shedding light on how they work as model size increases.
- CRaSh is a novel and effective strategy for improving the performance of OFT with billions of parameters.
- They demonstrate empirically that CRaSh improves performance across multiple datasets.

Cons:
- The paper will improve by clearly motivating how OFT ensures privacy. Adding the answers provided by the authors during rebuttal will improve the paper.

---

### Decision · Program_Chairs · 2023-10-07

**Decision:**

Accept-Main

**Comment:**

Offsite-Tuning (OFT) transfers transformer blocks between LLMs and downstream emulators to address this issue of privacy when instruction tuning LLMs with private data. The authors conducted an empirical study of LLMs, focusing on their representation and functional similarity. They discovered that as the model size grows, a unique modular structure emerges within LLM layers. They proposed a training-free strategy, CRaSh (Clustering, Removing, and Sharing), to derive improved emulators from LLMs that significantly enhances  the performance of OFT with billions of parameters.


Pros:
- This paper analyzes LLMs from the perspective of representation and functional similarity, shedding light on how they work as model size increases.
- CRaSh is a novel and effective strategy for improving the performance of OFT with billions of parameters.
- They demonstrate empirically that CRaSh improves performance across multiple datasets.

Cons:
- The paper will improve by clearly motivating how OFT ensures privacy. Adding the answers provided by the authors during rebuttal will improve the paper.